

# LiDAR-enhanced Closed-Loop Active Helix Approach

Zekai Chen, Aemilius A. W. van Vondelen, and Jan-Willem van Wingerden

Delft Center for Systems and Control, Delft University of Technology, Mekelweg 2, 2628CN Delft, the Netherlands

**Correspondence:** Zekai Chen (chenzk0429@gmail.com)

**Abstract.** The Helix approach has shown potential in increasing wind farm power production through enhancing wake mixing. By applying periodic blade pitch signals to upstream turbines, a helical wake is generated, which reduces velocity deficits for downstream turbines and mitigates the wake effect. While promising, the closed-loop implementation of the Helix approach remains largely unexplored, which could enable handling uncertainties and model errors in wind farm applications. This work presents a framework that integrates LiDAR-based wake measurements to enable such closed-loop control. First, a downwind-facing continuous-wave LiDAR is used to extract the hub vortex as the controlled variable. Second, we developed a control algorithm that regulates the hub vortex position in the Helix frame, thereby controlling the helical wake. Simulations in QBlade show that the framework enables a real-time, flow-informed closed-loop wake mixing approach. Compared with the open-loop cases, the framework corrects the shear-induced steady-state wake bias and enables measurement-informed, dynamic pitch adjustments under turbulence. In shear, bias correction increases downstream power but raises structural loads on both turbines; under turbulence, dynamic pitch control delivers a modest farm-level power gain with only minor load increases. These outcomes highlight the promise of flow-informed, closed-loop wake-mixing control and motivate further investigation.

## 1 Introduction

Wind energy plays a key role in mitigating climate change and achieving energy sustainability. However, in a wind farm, aerodynamic interactions between turbines reduce power production, increase structural loading and maintenance, shorten the lifetime of downstream turbines, and ultimately increase the levelized cost of energy (Houck, 2022). This interaction is called the "wake effect", referring to the reduced wind speed and increased turbulence intensity that the downstream turbine experiences because of the upstream turbines' wake.

To mitigate the negative influences of wakes on the downstream turbines, some research aims at arranging wind turbines more effectively (Kusiak and Song, 2010), while other studies are working on control methods to get the best performance out of wind farms. These control approaches, known as wind farm flow control strategies, involve the coordinated control of individual turbines to actively manipulate the wake flow. The objective is to enhance overall performance metrics of the wind farm, such as total power output, system lifespan, or levelized cost of energy (Meyers et al., 2022). In general, three categories of solutions have been proposed, such as the axial induction control method proposed by Annoni et al. (2016), which involves deliberately operating the upstream turbine at less than its maximum capacity, with the aim of leaving more energy in the wake for downstream turbines. However, the potential for increased energy extraction from static induction control is rather low,





making this method more suitable for load balancing within wind farms rather than overall production optimization (van der Hoek et al., 2019).

An alternative solution is wake steering, which refers to diverting the wake flow to mitigate the impact of the wake effect experienced by turbines downstream through yawing or tilting the upstream turbines. The redirected wake diverges from its initial path, decreasing its overlap with the rotor of a downstream turbine. As a result, the downstream rotor encounters higher speed and less turbulent wind, which can lead to an increase in power generation.

A different approach is suggested by Goit and Meyers (2015), where the wake is reduced by enhancing the mixing of the wake with the ambient free-stream air through dynamic variation of the induction. By promoting such mixing, the wake recovers energy more rapidly than through natural recovery alone. One implementation of this method is done by pitching periodically, hence creating a periodic structure in the wake, see Frederik et al. (2020b). Due to the periodic structure, this approach is more commonly referred to as the pulse approach. While this technique demonstrates substantial power gains in a two-turbine setup, it also leads to significant load increases due to variations in thrust force (Frederik and van Wingerden, 2022). Consequently, an alternative actuation method is proposed by Frederik et al. (2020a), where the position of the thrust force is rotated around its nominal axis rather than varying its magnitude. This generates a helical pattern in the wake, from which the approach derives its name as the Helix approach. This approach significantly reduces power fluctuations while also achieving better overall performance than the pulse approach. The Helix approach has attracted growing interest in the field, supported by large-eddy simulations (LES) and wind tunnel experiments (**?**van der Hoek, den Abbeele, Simao Ferreira, and van Wingerden, 2024), both demonstrating promising power gains.

Currently, the Helix approach is implemented in an open-loop configuration, offering the advantage of being fast and easy to implement. However, the absence of feedback information regarding the output wake limits the system's ability to dynamically adjust control strategies in the presence of uncertainties and model errors. For instance, a constant bias in the output may arise from external wind conditions or unmodeled system dynamics. Robust feedback control can address these challenges by accommodating these uncertainties in wind energy production (Meyers et al., 2022). Enabling such control requires the measurement of the output, namely, the wake, which, from a control perspective, corresponds to integrating a feedback mechanism into the control architecture. To access the wake information generated from the upstream turbine, the current work of Kerssemakers (2022) has investigated the use of blade root bending moments from the downstream turbine. Our work explores an alternative way to integrate wake measurement into control by using Light Detection and Ranging (LiDAR) sensing technology. When positioned downwind, a LiDAR can capture the wake generated by the upstream turbine, providing real-time feedback that enables the implementation of effective closed-loop control strategies. The work of Raach et al. (2017) utilizes this approach for closed-loop wake steering control, where a nacelle-based LiDAR system facing downwind is used to estimate the wake center, and a control system is designed to steer the wake into a desired position. Simulation result shows an approximately $4.5\%$ increase in total power output for a two-turbine wind farm compared to the open-loop approach (Raach et al., 2016).

To the best of the authors' knowledge, closed-loop wake mixing control based on LiDAR measurements remains unexplored. Inspired by the work of Raach et al. (2017), this paper aims to develop and implement a closed-loop wake mixing framework.





Among the two current wake mixing methods, this paper focuses on the Helix approach due to the better mixing and reduced tower loads and power fluctuations (Frederik et al., 2020a). Summarizing, the following contributions are presented in this work:

(1) We found an aerodynamic feature within the helical wake that exhibits a strong correlation with the wake dynamics and can be used for closed-loop wake mixing control.

   (2) We design and implement a framework, consisting of LiDAR and Control subsystems, for the Helix approach, achieving a flow-informed closed-loop implementation.

   (3) We evaluate (2) in a two-turbine setup in a free-vortex simulation platform (QBlade) simulation and compare it to the
traditional open-loop framework.

The remainder of this paper is organized as follows. Section 2 introduces preliminary knowledge, after which Section 3 presents the main contributions: the design of the framework, the supporting data processing pipelines, and the designed $\mathcal{H}_\infty$ controller tuned based on an identified model. Section 4 describes the simulation setup and test cases, followed by a presentation and analysis of the corresponding results. Lastly, conclusions are drawn in Section 5.

## 75  2   Prelimary Knowledge

In this section, a brief introduction is given to LiDAR, the Helix approach, and the simulation platform, which is essential background knowledge for understanding the proposed framework and the corresponding design.

### 2.1   LiDAR Sensing

LiDAR (Light Detection and Ranging) is a remote method for measuring wind speed that has gained attention in the wind
energy industry in recent years. It enables additional wake measurements to be incorporated into wind turbine controllers, thereby facilitating the development of advanced control strategies (Scholbrock et al., 2016). A LiDAR measures the wind speed based on the "Doppler Effect" with different scanning configurations. Figure 1 illustrates the working mechanism of a single-beam measurement device: the sent and reflected wavelength is compared, and the Doppler effect is used to derive the wind speed (Mikkelsen, 2014). The same analysis can be expanded to a multi-beam device; the main difference is that
measurements along the beam are also taken into consideration by a weight function. A primary limitation of wind LiDAR systems is that they measure only the component of the wind velocity along the laser beam's direction, referred to as the line-of-sight (LOS) velocity, denoted by $u_{\text{LOS}}$ (Raach et al., 2017).

The scanning configuration of a LiDAR refers to how the laser beams scan the space to get information. There are two types of LiDAR applied in the field of WFFC:

(1) Continuous-wave LiDAR, which shoots a continuous beam of light into the atmosphere, focusing at a predetermined distance ahead. Hence, this type of LiDAR only measures the wind field information at a specific distance.



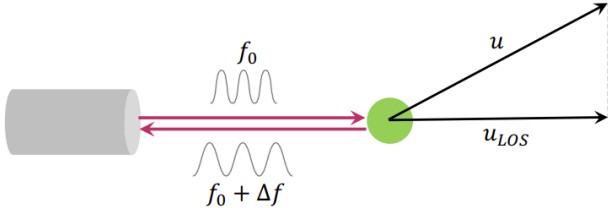

**Figure 1.** A simplified illustration that demonstrates the working mechanism of a single-beam measurement based on the Doppler effect. Here, $f_0$ and $f_0 + \Delta f$ denote the frequency of the sent and received waves, $u$ denotes the wind velocity, and $u_{\text{LOS}}$ denotes the line-of-sight component of the wind velocity.

(2) Pulsed LiDAR uses a timing-based method that waits for the reflected light to return at different times after a pulse of light is emitted from the LiDAR. This pattern enables the measurement of wind speeds at various distances.

The work of Köpp et al. (2005) compared wake–vortex measurement quality between continuous-wave (CW) and pulsed LiDAR and found them to be nearly equivalent. The key distinction is that pulsed LiDAR samples multiple ranges at different return times, enabling analysis of the temporal evolution of vortex circulation, whereas CW LiDAR measures at a single focal distance, so all measurements share the same return time. In this work, we adopt CW LiDAR because the uniform return time of all measurements simplifies controller design, and wake evolution is not our primary focus. Nevertheless, pulsed LiDAR warrants further investigation when wake evolution is of primary interest.

## 2.2 The Helix Approach

The Helix approach generates a helical wake by applying individual sinusoidal pitch signals to each blade, resulting in a directional moment on the rotor. This moment exerts a periodic force on the airflow, continuously steering the wake direction (Frederik et al., 2020a). Normally, the dynamics of wind turbine rotor blades are expressed in the rotating frame attached to the individual blades. The rotor, however, responds as a whole in the fixed frame. As a result, the multi-blade coordinate transform (MBC) is used to integrate the dynamics of individual blades and express them in a fixed frame, as Eq. 1 shows

$$\begin{bmatrix} \beta_{\text{col}} \\ \beta_{\text{tilt}} \\ \beta_{\text{yaw}} \end{bmatrix} = \frac{2}{3} \cdot \underbrace{\begin{bmatrix} 1/2 & 1/2 & 1/2 \\ \cos\psi_1 & \cos\psi_2 & \cos\psi_3 \\ \sin\psi_1 & \sin\psi_2 & \sin\psi_3 \end{bmatrix}}_{T(\omega_r t)} \cdot \begin{bmatrix} \beta_1 \\ \beta_2 \\ \beta_3 \end{bmatrix}. \tag{1}$$

In Eq. 1, $\psi_i$ represents the azimuth angle of blade $i$, $\beta_{\text{col}}$ represents the collective pitch signal, and $\beta_{\text{tilt}}$ and $\beta_{\text{yaw}}$ denote the fixed frame and azimuth-independent tilt and yaw pitch signal, respectively. Conversely, the pitch angle of individual blades $\beta_i$ can be acquired based on the collective, tilt, and yaw pitch signal of the rotor in the fixed frame by inverse MBC transformation:

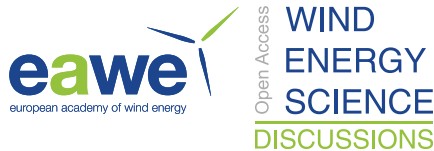

$$
\quad \begin{bmatrix} \beta_1 \\ \beta_2 \\ \beta_3 \end{bmatrix} = \underbrace{\begin{bmatrix} 1 & \cos\left(\psi_1 + \psi_{\text{off}}\right) & \sin\left(\psi_1 + \psi_{\text{off}}\right) \\ 1 & \cos\left(\psi_2 + \psi_{\text{off}}\right) & \sin\left(\psi_2 + \psi_{\text{off}}\right) \\ 1 & \cos\left(\psi_3 + \psi_{\text{off}}\right) & \sin\left(\psi_3 + \psi_{\text{off}}\right) \end{bmatrix}}_{T^{-1}(\omega_r t + \psi_{\text{off}})} \cdot \begin{bmatrix} \beta_{\text{col}} \\ \beta_{\text{tilt}} \\ \beta_{\text{yaw}} \end{bmatrix}, \quad (2)
$$

where $\psi_{\text{off}}$ represents an azimuth offset that compensates for unmodelled actuator delays and blade flexibility, which is essential for achieving full decoupling of the tilt and yaw channel (Mulders et al., 2019).

In practice, the Helix approach is implemented by applying sinusoidal signals to the tilt ($\beta_{\text{tilt}}$) and yaw ($\beta_{\text{yaw}}$) angles. The frequency at which these signals are varied is characterized by the non-dimensional Strouhal number St:

$$
\quad \text{St} = \frac{f_e D}{V_\infty}, \quad (3)
$$

where $f_e$ is the excitation frequency of the tilt and yaw commands, $D$ is the rotor diameter, and $U_\infty$ is the free stream wind velocity. Strouhal values are generally selected between 0.2 and 0.4 as recommended by previous work (Frederik, Weber, Cacciola, Campagnolo, Croce, Bottasso, and van Wingerden, 2020b; Frederik, Doekemeijer, Mulders, and van Wingerden, 2020a). This leads to the tilt and yaw pitch commands for Helix wake mixing as the following equation shows:

$$
\quad \begin{bmatrix} \beta_{\text{tilt}} \\ \beta_{\text{yaw}} \end{bmatrix} = \begin{bmatrix} A\sin\left(\omega_e t\right) \\ A\sin\left(\omega_e t \pm \pi/2\right) \end{bmatrix}, \quad (4)
$$

where $A$ is the amplitude of Helix excitation, usually no larger than a few degrees due to practical constraints such as pitch rate limitations (Taschner et al., 2023), and $\omega_e = f_e 2\pi$.

Two Helix variants are distinguished by a phase difference of $+\pi/2$ and $-\pi/2$ between the tilt and yaw pitch signals, resulting in a clockwise (CW) and counterclockwise (CCW) Helix, respectively. While the actuation frequency in the fixed frame remains identical for both variants, the actual frequency applied by the pitch actuator varies once the tilt and yaw control commands are mapped to the rotating frame. This mapping leads to a Helix frequency in the rotating frame of either $\omega_r \pm \omega_e$ ($1P \pm f_e$), depending on whether the Helix is CW or CCW. Generally, a CCW Helix results in higher farm-level energy gains (Taschner et al., 2023), while the CW Helix is favored for lower damage to the pitch bearing (van Vondelen et al., 2023), which can be explained by the lower effective actuation frequency of $1P - f_e$. In this work, the CCW Helix is selected due to better energy gain.

## 2.3 Simulation Tools

This study employs the NREL 5MW wind turbine as the object of study, see Jonkman et al. (2009) for details. This turbine is widely used in wind energy research, offering a well-established benchmark. All simulations are conducted using QBlade (Marten et al., 2013), which uses a free-wake vortex method to simulate the flow field and the wake around the turbine. This method is known for its accuracy in the near wake and for being computationally more efficient than the





LES method (Shaler et al., 2020). Although free-wake vortex methods may suffer from numerical instabilities in the far wake (van den Berg et al., 2023), this limitation is not critical for this study, as the downstream turbine is positioned within the near to mid-wake region. This placement is sufficient to capture relevant wake dynamics, as demonstrated in Marten et al. (2020).

The QBlade setup for aerodynamic simulation in this work is chosen under the principle of finding a trade-off between computational time and accuracy. Both wake modeling and the vortex modeling settings influence this balance: the former directly regulates the number of elements in the wake, while the latter are settings that influence vortex performance. The simulation settings used in this work follow those in (van den Berg et al., 2023), as the two-turbine configurations are identical.

## 3    Closed-loop Active Wake Mixing Framework

This section serves as the core contribution of this paper: the proposed structure and design of the closed-loop active wake mixing framework. The following section focuses on the overall framework structure, followed by the design details about the LiDAR and control subsystems.

### 3.1    Overall Framework Stucture

To enable LiDAR-based closed-loop wake mixing control within a wind farm, two main tasks must be considered: (1) the
measurement task, and (2) the control task. Thus, the overall system is designed to have two subsystems, each dedicated to fulfilling one of these tasks:

(1) **The LiDAR Subsystem** consists of a LiDAR facing downwind and a supporting pipeline for data processing. The design should fulfill two functionalities in real-time:

- Helical wake data sampling.
- Helical wake feature acquisition.

(2) **The Control Subsystem** consists of a controller and the supporting components for closed-loop control. The design should fulfill the functionality of:

- Generate individual blade pitch inputs $\beta_i$ based on real-time flow measurements.
- Correct the helical wake based on the current output of the system and the given reference, compensating for any
detected misalignment.

Consequently, the block diagram of the overall system is constructed as shown in Fig. 2. The design of the overall framework structure is inspired by the work of Raach et al. (2017).





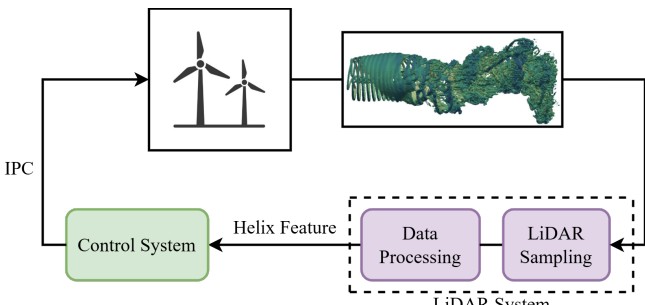

**Figure 2.** The diagram of the overall closed-loop control system consisting of the LiDAR subsystem and the Control subsystem.

### 3.2 LiDAR Subsystem Design

To achieve the aforementioned functionalities, the LiDAR configuration, supporting assumptions, and modeling approach are
first presented. A feature for control is then selected. Finally, a coordinate transformation similar to the MBC transform is
introduced to simplify the controller design.

#### 3.2.1 LiDAR Setup and Modeling

In this work, several assumptions regarding the LiDAR are made as follows:

(1) The half cone angle $\varphi$ of the LiDAR is configured to encompass the information of the entire plane with a diameter the
same as the rotor disk $D$.

(2) The focal distance of the LiDAR is the same as the diameter of the rotor disk $D$.

(3) A plane can be considered as a collection of many points; when a LiDAR measures information about a plane, it effectively samples data from these individual points by emitting multiple laser beams. Furthermore, it is assumed that these
laser beams are emitted simultaneously with no phase delay.

The LiDAR measurement can be modeled by a point measurement in the wind field (Raach et al., 2017). In the inertial
coordinate system, this is done by projecting the wind vector in three direction $u = \begin{bmatrix} u_{i,x} & u_{i,y} & u_{i,z} \end{bmatrix}$ onto the normalized
laser vector in the $i$th point $\begin{bmatrix} x_i & y_i & z_i \end{bmatrix}$ with focus distance $f_i = \sqrt{x_i^2 + y_i^2 + z_i^2}$ by:

$$u_{\mathrm{LOS},i} = \frac{x_i}{f_i} u_{i,x} + \frac{y_i}{f_i} u_{i,y} + \frac{z_i}{f_i} u_{i,z} \tag{5}$$

Figure 3 shows a 3-dimensional view of the LiDAR setup: a LiDAR is mounted on the top of the wind turbine nacelle,
orienting downwind. The LiDAR measures the wind speed information of a plane with the same diameter as the rotor disk
with a focal distance of $1D$. This distance is chosen as a trade-off between wake data quality, measurement feasibility, and the
upper bandwidth limit of the controller imposed by output delay, further discussed in Section 3.3.



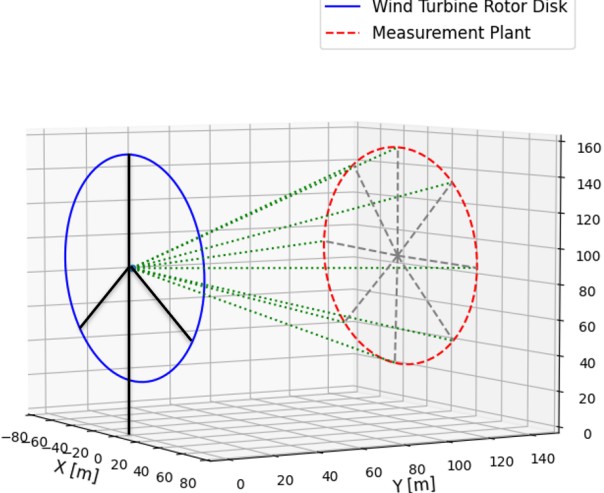

**Figure 3.** The 3-Dimensional view of the LiDAR Setup.

### 3.2.2 Helix Feature Extraction

This section identifies a suitable controlled variable in the helical wake experimentally by analyzing the LiDAR-sampled data
in simulations with the Helix approach being activated.

Figure 4 presents LiDAR sampling snapshots in QBlade over a complete Helix cycle $T_e$. A counter-clockwise (CCW) Helix
is applied with a Strouhal number of $\mathrm{St} = 0.3$, a uniform inflow wind speed of $10\,\mathrm{m/s}$, and a Helix amplitude of 3 degrees.
In the figure, wind speed is visualized through a color map, where higher velocities correspond to brighter shades. A distinct
high-velocity region is observed rotating at the excitation frequency $f_e$, corresponding to the hub vortex of the wind turbine.
This structure extends along the streamwise direction and has been previously documented in Iungo et al. (2013). The findings
of Coquelet et al. (2023) further confirm the existence and rotational behavior of the hub vortex in helical wakes. Addition-
ally, we analyzed sampled data under varying wind conditions, including vertical shear (exponential factor 0.2), turbulence
(intensity 6%), and their combination, while maintaining an average wind speed of $10\,\mathrm{m/s}$. In all cases, the hub vortex remains
distinguishable and retains its rotational behavior.

Experiments were conducted by randomly varying the Helix amplitude between 1 and 6 degrees and the Strouhal number
between 0.1 and 0.4 while recording the rotation magnitude and frequency of the hub vortex. These parameter ranges were
selected as the regime in which the Helix is most effective, see Taschner et al. (2023). The correlation between the hub vortex
rotation magnitude and the Helix amplitude was found to be 0.9987, and the correlation between the hub vortex rotation
frequency and the Strouhal number was 0.9995. These results show that Helix amplitude changes directly affect hub vortex
rotation magnitude, while Strouhal number changes affect rotation frequency. This strong correlation indicates that controlling





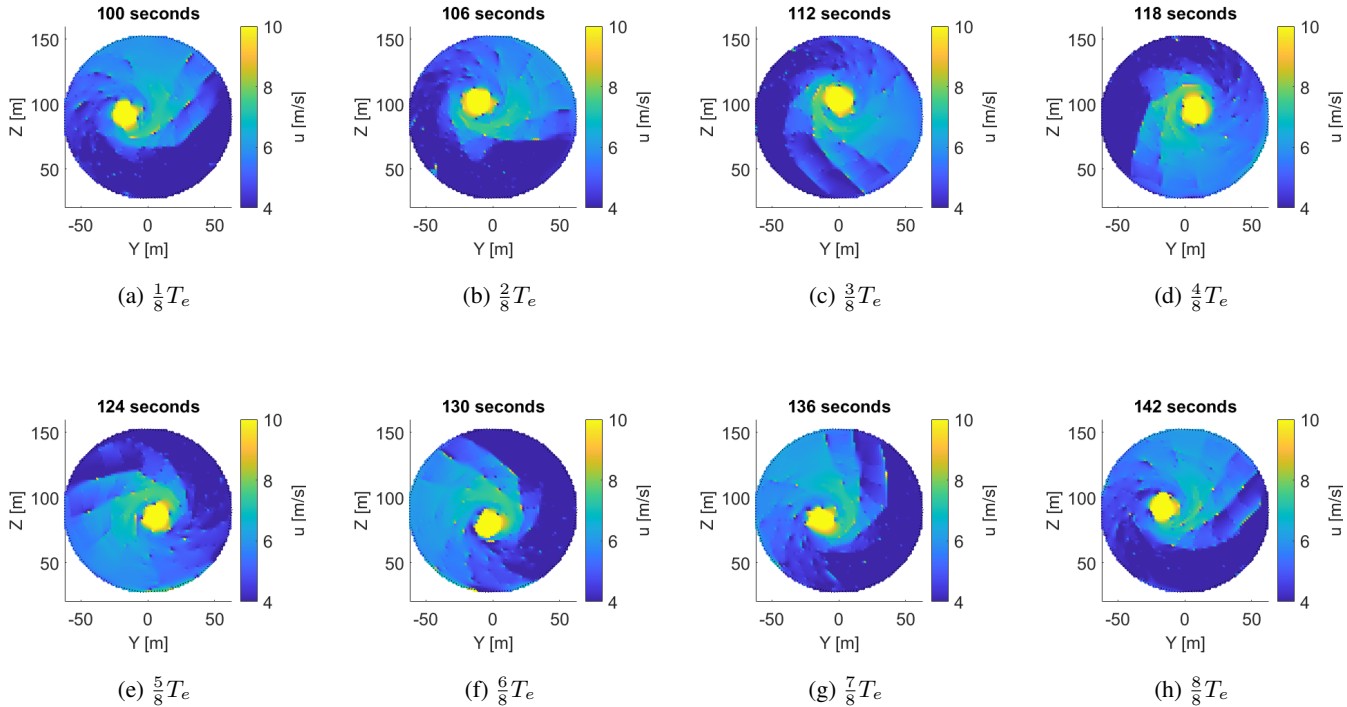

**Figure 4.** LiDAR sampling snapshots over a complete Helix cycle $T_e$.

**Table 1.** The frequency of helical wake in different coordinate frames.

| Coordinate Frame | Frequency |
|---|---|
| Rotating | $1P \pm f_e$ |
| Fixed | $f_e$ |
| Helix | $DC$ |

the hub vortex is equivalent to controlling the helical wake. Additionally, the hub vortex is readily distinguishable through signal filtering due to its elevated velocity. It is therefore selected as the controlled variable of the helical wake.

### 3.2.3 Helix Frame Transformation

A noticeable characteristic of the hub vortex is its continuous rotation relative to the fixed frame at the Helix frequency $f_e$. To simplify the controller design, the Helix frame transformation proposed by van Vondelen et al. (2024) is employed to map the hub vortex from the fixed frame to the Helix frame, in which the vortex appears relatively stationary.



To achieve this, the principle of modulation-demodulation is applied, transforming the rotating Helix from the rotating coordinate frame to the Helix coordinate frame, shifting the $1P + f_e$ Helix rotation (assuming a CCW Helix) to the DC gain. The summary of the frequency of the Helix in different coordinate frames is shown in Table 1. The derivation of the Helix frame transform resembles the MBC transform. The main difference is that the excitation frequency $\omega_e$ is included:

$$\begin{bmatrix} \beta_{\text{col}}^e \\ \beta_{\text{tilt}}^e \\ \beta_{\text{yaw}}^e \end{bmatrix} = T_{\text{cm}}(\omega_e t) \begin{bmatrix} \beta_1 \\ \beta_2 \\ \beta_3 \end{bmatrix}, \tag{6}$$

where

$$T_{\text{cm}}(\omega_e t) = \frac{2}{3} \begin{bmatrix} 1/2 & 1/2 & 1/2 \\ \cos(\omega_1) & \cos(\omega_2) & \cos(\omega_3) \\ \sin(\omega_1) & \sin(\omega_2) & \sin(\omega_3) \end{bmatrix}, \tag{7}$$

and $\omega_i = \psi_i + \omega_e t$, representing the CCW Helix frequency.

For ease of implementation, it is useful to decouple the transform $T(\omega_r t + \omega_e t)$ as shown in van Vondelen et al. (2024). This process starts with the sum of the angles by using the angle sum identity matrix as Eq. 8 shows:

$$\begin{bmatrix} \cos(\omega_r t + \omega_e t) \\ \sin(\omega_r t + \omega_e t) \end{bmatrix} = \begin{bmatrix} \cos(\omega_e t) & -\sin(\omega_e t) \\ \sin(\omega_e t) & \cos(\omega_e t) \end{bmatrix} \cdot \begin{bmatrix} \cos(\omega_r t) \\ \sin(\omega_r t) \end{bmatrix}. \tag{8}$$

Subsequently, Eq. 6 can be rewritten as Eq. 9 with $R(\omega_e t)$ being the rotation matrix:

$$\begin{bmatrix} \beta_{\text{col}}^e \\ \beta_{\text{tilt}}^e \\ \beta_{\text{yaw}}^e \end{bmatrix} = \underbrace{\begin{bmatrix} 1 & 0 & 0 \\ 0 & \cos(\omega_e t) & -\sin(\omega_e t) \\ 0 & \sin(\omega_e t) & \cos(\omega_e t) \end{bmatrix}}_{R(\omega_e t)}$$

$$\times \underbrace{\frac{2}{3} \cdot \begin{bmatrix} 1/2 & 1/2 & 1/2 \\ \cos\psi_1 & \cos\psi_2 & \cos\psi_3 \\ \sin\psi_1 & \sin\psi_2 & \sin\psi_3 \end{bmatrix}}_{T(\omega_r t)} \begin{bmatrix} \beta_1 \\ \beta_2 \\ \beta_3 \end{bmatrix}. \tag{9}$$

Consequently, the Helix frame transform is implemented simply by multiplying a rotation matrix $R(\omega_e t)$ by the MBC transform, allowing the original non-rotating frame to rotate at a frequency of $\omega_e$ rad/s, thereby making the previously rotating hub vortex stationary in the Helix frame. Conversely, the inverse Helix frame transformation can be derived as Eq. 10 shows, noting that the azimuth offset $\psi_{\text{off}}$ is added to decouple the tilt and yaw channels (Mulders et al., 2019):



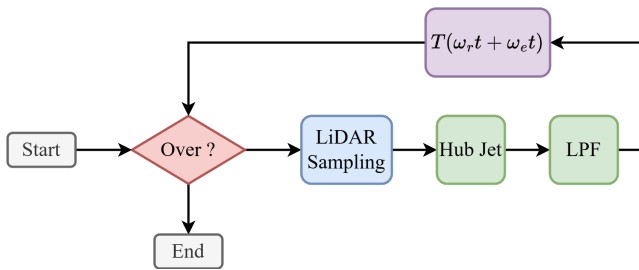

**Figure 5.** Flow chart of the LiDAR data processing pipeline.

$$
\quad
\begin{bmatrix} \beta_1 \\ \beta_2 \\ \beta_3 \end{bmatrix} =
\underbrace{\begin{bmatrix}
1 & \cos(\psi_1 + \psi_{\mathrm{off}}) & \sin(\psi_1 + \psi_{\mathrm{off}}) \\
1 & \cos(\psi_2 + \psi_{\mathrm{off}}) & \sin(\psi_2 + \psi_{\mathrm{off}}) \\
1 & \cos(\psi_3 + \psi_{\mathrm{off}}) & \sin(\psi_3 + \psi_{\mathrm{off}})
\end{bmatrix}}_{T^{-1}(\omega_r t + \psi_{\mathrm{off}})}
$$

$$
\times \underbrace{\begin{bmatrix}
1 & 0 & 0 \\
0 & \cos(\omega_e t) & \sin(\omega_e t) \\
0 & -\sin(\omega_e t) & \cos(\omega_e t)
\end{bmatrix}}_{R^{-1}(\omega_e t)} \cdot
\begin{bmatrix} \beta_{\mathrm{col}}^e \\ \beta_{\mathrm{tilt}}^e \\ \beta_{\mathrm{yaw}}^e \end{bmatrix}. \tag{10}
$$

A similar operation can be applied to the input signals of $\beta_{\mathrm{tilt}}$ and $\beta_{\mathrm{yaw}}$, converting to $\beta_{\mathrm{tilt}}^e$ and $\beta_{\mathrm{yaw}}^e$. Therefore, the original MIMO system $(\beta_{\mathrm{tilt}}, \beta_{\mathrm{yaw}}) \rightarrow (z, y)$ is mapped to $(\beta_{\mathrm{tilt}}^e, \beta_{\mathrm{yaw}}^e) \rightarrow (z^e, y^e)$, offering a more static signal representation. Moreover, this transformation reduces system coupling, as evidenced by the dominant diagonal elements in the Relative Gain

Array (RGA), see Skogestad and Postlethwaite (2005), thereby simplifying control design. Note that mean centering needs to be applied to signals to eliminate extra oscillating components. As a result, the overall LiDAR processing pipeline is developed as Fig. 5 shows.

The pipeline consists of three parts:

(1) **LiDAR Sampling**: Capture wind speed data $u_{\mathrm{LOS}}$ at the specified focal distance of 1D downwind.

(2) **Data Processing**:

    – **Hub Jet Filter**: Extract the coordinates of the hub vortex in the rotating frame, $(y, z)$, by isolating the high-speed region and averaging the positions of the filtered points.

    – **Low Pass Filter**: Remove high-frequency noise from the signal. The finite impulse response (FIR) filter is chosen for its desirable properties like guaranteed stability, absence of limit cycles, and linear phase (Neuvo et al., 1984).

The filter order is selected as 50 in the trade-off between phase delay and filtering performance, and the cut-off frequency of the low-pass filter is chosen as $0.05\,\mathrm{Hz}$ since the Helix is in the frequency of $0.0238\,\mathrm{Hz}$.



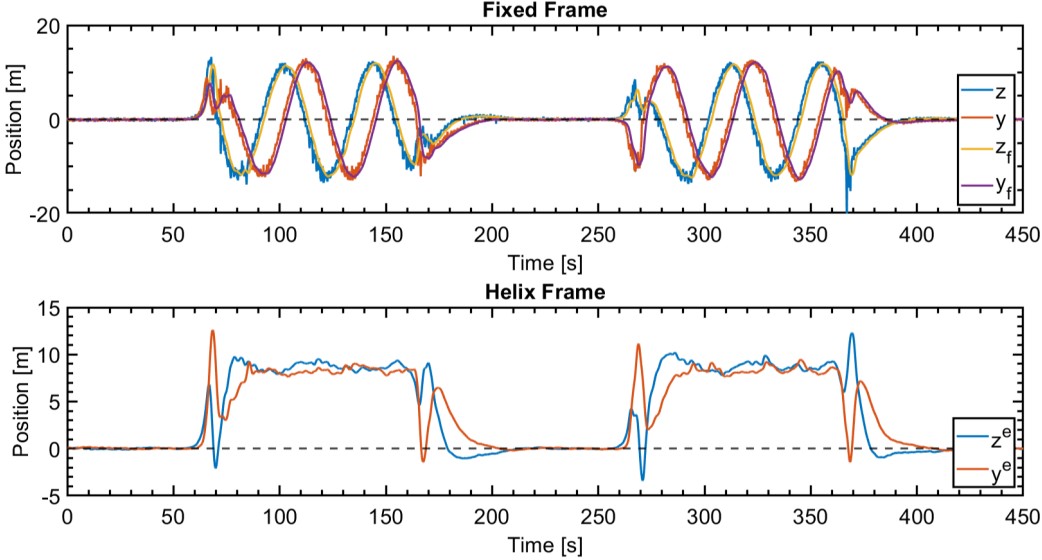

**Figure 6.** Output of the data processing pipeline. The top panel shows hub vortex signals in the fixed frame; the bottom panel shows them in the Helix frame. Here, $(y, z)$ are the original signals, $(y_f, z_f)$ the filtered ones, and $(y^e, z^e)$ the results after applying the Helix frame transform.

(3) **Helix Frame Transform**: Map the hub vortex from the fixed frame to the Helix frame where it becomes static.

Figure 6 presents the output of the data processing pipeline. The Helix approach, configured with $\mathrm{St} = 0.3$ and $u = 10\,\mathrm{m/s}$, is activated during two discrete intervals: 50 to 150 seconds and from 250 to 350 seconds. Outside these periods, the wind turbine operates under baseline conditions. The result confirms that the pipeline effectively maps the rotating signals from the fixed frame to the Helix frame. Although the signal in the Helix frame is not fully static, likely due to residual noise that the low-pass filter cannot fully eliminate, a clear trend toward a constant value is still observable.

## 3.3 Control Subsystem Design

The main challenge of designing the control system is the presence of a time delay $\tau$, as the wake needs to take time to travel to the measurement location. This delay $\tau$ is categorized as the output delay, defined as the delay between the time the system state or output changes and the time this change is observed (Zhang and Xie, 2007). For control purposes, two assumptions are made for the delay $\tau$ in this work:

(1) The value of $\tau$ is assumed to align with Taylor's frozen turbulence hypothesis (Taylor, 1938) as introduced by Scholbrock et al. (2016), which states that the value of delay is only determined by the measurement distance $x$ and the average inflow wind speed $u_{\mathrm{in}}$:

$$\tau = \alpha \cdot \frac{x}{u_{\mathrm{in}}}. \tag{11}$$





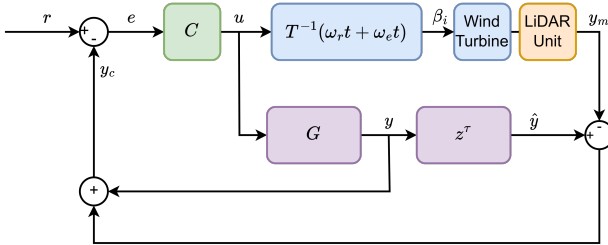

**Figure 7.** The block diagram of the control system.

The coefficient $\alpha$ is used for calibrating the value.

(2) The delay for each output $y_i$ to each input $u_j$ remains the same, since the Helix frame transform does not alter the physical advection time of the yaw and tilt components of the helical wake, which reach the measurement point simultaneously. As a result, the dynamic of the system in the Helix frame can be expressed as:

$$\begin{bmatrix} z^e \\ y^e \end{bmatrix} = \underbrace{\begin{bmatrix} G_{11}(z) & G_{12}(z) \\ G_{21}(z) & G_{22}(z) \end{bmatrix} z^{-\tau}}_{\mathbf{G}} \begin{bmatrix} \beta^e_{\text{tilt}} \\ \beta^e_{\text{yaw}} \end{bmatrix}. \tag{12}$$

In this work, the average inflow wind speed of every case is set to be $10\,\text{m/s}$. Accordingly, a consistent delay time is found as $T = 11.2$ seconds via Padé approximation.

To control a delayed system, the Smith predictor based on internal model control (IMC) is adopted, as it effectively compensates for time-invariant delays. This approach is well-suited for systems with known constant dead-time, as demonstrated in Abe and Yamanaka (2003). Figure 7 shows the general concept of the controller. The reference signal $r$ is the desired coordinate of the hub vortex in the Helix frame, represented by $[z^e_r, y^e_r]^T$. The control input $u$ denotes the tilt and yaw pitch signal in the Helix frame, expressed as $[\beta^e_{\text{tilt}}, \beta^e_{\text{yaw}}]^T$. Since the wind turbine cannot directly receive tilt and yaw signals, the inverse Helix frame transformation $T^{-1}(\omega_r t + \omega_e t)$ is applied to convert the inputs into the blade pitch signals $\beta_i$. Moreover, the LiDAR unit samples the wind turbine output and converts the hub vortex into the Helix frame.

### 3.3.1 Internal Model Identification

The presented controller follows the idea of internal model control, in which the difference between the actual system output and a predicted output is used within the controller to regulate the system (Raach et al., 2017). Therefore, a model $G$ that describes the dynamics of the helical wake effect, or the dynamics of $(\beta^e_{\text{tilt}}, \beta^e_{\text{yaw}}) \rightarrow (z^e, y^e)$, is essential. In this work, the internal model $G$ is acquired through system identification of the experimental data, focusing on wind speed $u_{\text{in}} = 10\,\text{m/s}$.

To identify a model, the Pseudo-Random Binary Noise (PRBN) with a magnitude of 1 degree, and filtered by a bandpass filter between a frequency range of $[0, 0.03]$ Hz to ensure compatibility with the actuator's bandwidth, is selected as the excitation signal due to its effectiveness in exciting a broad spectrum of system frequencies, facilitating a comprehensive capture of the





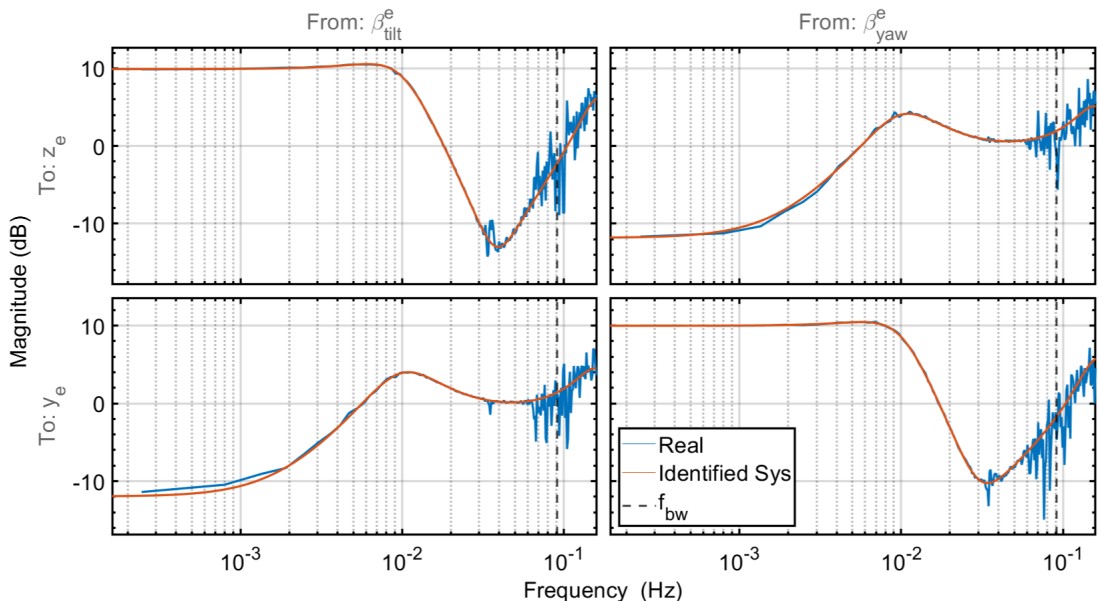

**Figure 8.** Comparison of the PBSID-opt identified model (orange) against the spectral averaged input and output data (blue). The dashed line indicates the estimated bandwidth frequency, which defines the frequency of interest.

system's dynamic characteristics (Godfrey, 1991). Furthermore, the optimized predictor-based subspace identification (PBSID-
opt) is used. This method is based on the well-established stochastic subspace identification approach, which uses input-output data to estimate a linear model by persistently exciting the system with an input signal containing a wide range of frequencies (van der Veen et al., 2013). The sizes of past and future windows are set identically to 200 to achieve a balance among computational speed, noise sensitivity, and accuracy. Analysis of the singular values produced by the PBSID-opt method shows a noticeable drop beyond order 4, indicating that a model of order 4 appropriately captures the spectral characteristics
of the input-output data, as can be observed in Fig. 8. This order offers a balance between model fidelity and computational complexity. An azimuth offset of $\psi_{\mathrm{off}} = 6$ degrees is applied to facilitate decoupling, as it yields diagonal elements of the RGA matrix close to 1, indicating a well-decoupled system.

The result of Fig. 8 yields several observations:

(1) The identified system successfully captures the system dynamics within the frequency range.

(2) The difference in steady-state magnitude between the diagonal and off-diagonal transfer functions indicates a degree of decoupling within the system. Specifically, the steady-state gain of $G_{11}$ and $G_{22}$ are positive while those of $G_{12}$ and $G_{21}$ are negative. This implies that $\beta_{\mathrm{tilt}}^{e}$ influences $z^{e}$ and $\beta_{\mathrm{yaw}}^{e}$ influences $y^{e}$ dominantly in steady-state frequencies. The steady-state RGA matrix of $\begin{bmatrix} 0.9935 & 0.0065 \\ 0.0065 & 0.9935 \end{bmatrix}$ supports this furtherly.



Since this research employs a linear system identification method, $G$ is linear time-invariant. Consequently, $G$ is only ap-
plicable within a specific operating range. Although the model $G$ exhibits steady-state decoupling, strong couplings at the
bandwidth frequency, supports by the RGA matrix at the bandwidth frequency of $\begin{bmatrix} -1.2780 & 2.2780 \\ 2.2780 & -1.2780 \end{bmatrix}$, complicate the
design of a decentralized controller combined with a pre-compensator, such as a diagonal control structure adopted in **?**. A
diagonal controller could be implemented by choosing a very low crossover frequency, but the slow reaction time would offer
limited benefit. Consequently, an $\mathcal{H}_\infty$ controller is adopted for its robustness to uncertainty and modeling errors in MIMO
systems.

### 3.3.2 $\mathcal{H}_\infty$ Controller Synthesis

The $\mathcal{H}_\infty$ controller synthesis uses the general control configuration, as Fig. 9 shows, where $P$ is the generalized plant and $K$
the generalized controller.

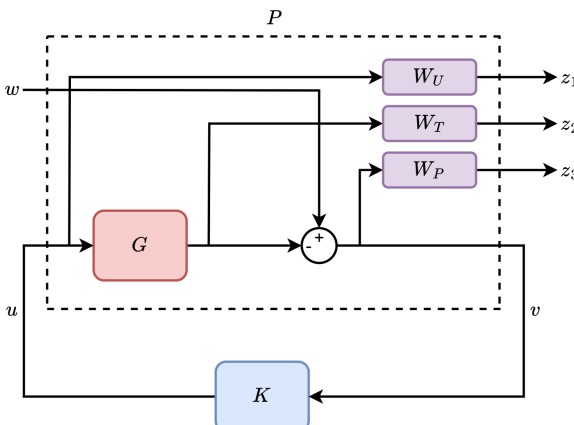

**Figure 9.** Generalized plant $P$ with performance signals $z_1$, $z_2$, $z_3$, and input $w$. Furthermore, $G$ denotes the identified model, $K$ represents
the controller, and $W_P$, $W_U$, and $W_T$ represent the performance weights.

The idea of formulating a general control problem is to find a controller $K$ to minimize the $\mathcal{H}_\infty$ norm of the transfer
function from input $w$ to performance output $z$, see Skogestad and Postlethwaite (2005). Weight functions $W_U$, $W_T$, and $W_P$
are integrated to provide different weighted performance measures as outputs $z_1$ to $z_3$. The $\mathcal{H}_\infty$ controller synthesis considers
three criteria to design and evaluate the performance of the controller: the sensitivity $S$, the complementary sensitivity $T$, and
the controller sensitivity $U$, defined as

$$S = \frac{1}{1+GK}, \ T = \frac{GK}{1+GK}, \ U = \frac{K}{1+GK}. \tag{13}$$

For the physical interpretation, $S$ gives the transfer function from the disturbance to the system output, $T$ is the transfer function
from the reference to the output, and $-U$ is the transfer function from the disturbance to the control signal.





As a result, the controller $K$ is obtained by solving the mixed-sensitivity optimization problem, as defined in Eq. 14:

$$\min_K \left\| \begin{matrix} W_P S \\ W_T T \\ W_U U \end{matrix} \right\|_\infty = \min_K \left\| \begin{matrix} W_P (1 + GK)^{-1} \\ W_T GK (1 + GK)^{-1} \\ W_U K (1 + GK)^{-1} \end{matrix} \right\|_\infty . \tag{14}$$

For the closed-loop system, a good disturbance rejection is desired, and therefore, $S$ should be small for low frequencies.
Furthermore, the control effort should be limited by having a roll-off in $T$ after the bandwidth. Based on this, the weight
functions are designed. $W_P(s)$ is designed as a low-pass filter with the form of:

$$W_P(s) = \frac{s/M + \omega_{\text{cl}}}{s + A\omega_{\text{cl}}}, \tag{15}$$

where $\omega_{\text{cl}}$ denotes the desired closed-loop bandwidth, $A$ is the desired disturbance attenuation inside the bandwidth, and $M$
the desired bound on $||S||_\infty$ and $||T||_\infty$. The upper bound $M$ and lower bound $A$ of $S$ are set to $M = 10$ and $A = 0.625$,
respectively. The desired closed-loop bandwidth $\omega_{\text{cl}}$ is set to $\omega_{\text{cl}} = 0.02\,\text{rad/s}$ due to the limitation introduced by the non-
minimum phase zero of $G(s)$. Specifically, the upper bound of $\omega_{\text{cl}}$ is selected to remain below $z/2$, where $z$ is the real part of
the non-minimum phase zeros in continuous time, as proven in Skogestad and Postlethwaite (2005).

The non-minimum phase zero indirectly limits the controller gains for the stability requirement. Consequently, the controller
weight function $W_U(s)$ is designed as a band-limited high-pass filter to attenuate the control input in the high frequency as
below:

$$W_U(s) = 0.4B^2 \cdot \frac{s^2 + \sqrt{2}\omega_c + \omega_c^2}{s^2 + B\sqrt{2}\omega_c s + (B\omega_c)^2}. \tag{16}$$

Parameter $B$ scales the frequency at which control effort starts to be limited, and $\omega_c$ is related to the cutoff frequency. In this
study, $B$ is selected as 10, and the crossover frequency $\omega_c$ is set to $0.15\,\text{rad/s}$ according to the pitch controller changing rate.
Finally, $W_T(z)$ is kept at 0 since we focused on tracking performance and disturbance rejection at low frequencies (Skogestad
and Postlethwaite, 2005).

## 4 Results and Analysis

This section presents the results of the proposed control framework, including an evaluation of its reference tracking perfor-
mance and its impact on power production and blade damage-equivalent loads (DEL) in both flapwise and edgewise directions.

### 4.1 Simulation Setup

The proposed framework is evaluated in QBlade. The controller is tested by comparing it with the open-loop controller in four
different wind conditions: uniform wind, shear, turbulence, and combined shear and turbulence.

A two-turbine wind farm is created for simulation, where the downstream turbine is placed 4 rotor diameters (4D) away
from the upstream turbine. This distance balances the trade-off between the QBlade simulation quality with realistic turbine



**Table 2.** Parameters of wind field generated by TurbSim.

| Description | Value |
| --- | --- |
| Reference Wind Speed | 10 m/s |
| Grid Width, Height | 179, 179 meters |
| Grid $Y$, $Z$ Points | 20, 20 |
| Turbine Class | I |
| IEC Standard | 61400-1Ed3 |
| Wind Type | NTM |
| Spectral Model | IECKAI |
| Horizontal & Vertical Inflow Angle | 0 deg |
| Roughness Length | 0.01 |
| Wind Profile Type | Power Law |
| Shear Reference Height | 90 meters |
| Jet Height | 100 meters |

spacing. The exponential factor of shear is set to $0.2$, and the turbulence intensity (TI) is set to $6\%$ according to the IEC 61400-1
design standard to mimic the usual condition offshore (Burton et al., 2011). In this work, the shear and turbulence are added
by using the Turbulence Simulator (TurbSim), see Jonkman et al. (2014). The primary parameters of the generated wind field
are summarized in Table 2.

To study the effect of the proposed control framework on power production and fatigue, 8 cases are performed as listed
in Table 3. For all cases, the average inflow wind speed is kept constant at $u_{in} = 10\,\text{m/s}$ and the Strouhal number is set to
$St = 0.3$. All helical wakes generated rotate in the CCW direction. Each simulation ran for 15 minutes, with the initial 300
seconds excluded from analysis to account for wake development and transient effects.

**4.2  Open-Loop Helix in Different Wind Conditions**

Figure 10 illustrates the hub vortex trajectory in the fixed frame of different wind conditions compared to the uniform wind
case. The figure reveals a consistent bias in the $z$-direction under shear conditions, increased oscillations with turbulence, and
the presence of both bias and oscillation when shear and turbulence are combined. These trajectory changes impact both power
production and fatigue loading, as Fig. 11 shows. The results reveal that:

(1) **Shear** When only shear is present, the power loss of both turbines is evident. This is consistent with the findings of Pari-
nam et al. (2023), which reports that a higher shear resulted in a reduction in wake recovery and a lower TI in the wake
as a whole, thereby increasing the power loss of turbines located downstream. Moreover, the cumulative blade edgewise
and flapwise DEL decreased.





**Table 3.** Overview of all test cases.

| Case | Wind | Control | WT$_1$ | WT$_2$ |
|------|------|---------|--------|--------|
| BL | Uniform | - | Greedy | Greedy |
| OL1 | Uniform | OL | CCW Helix | Greedy |
| CL1 | Uniform | CL | CCW Helix | Greedy |
| OL2 | Shear 0.2 | OL | CCW Helix | Greedy |
| CL2 | Shear 0.2 | CL | CCW Helix | Greedy |
| OL3 | TI 6% | OL | CCW Helix | Greedy |
| CL3 | TI 6% | CL | CCW Helix | Greedy |
| OL4 | Shear + TI 6% | OL | CCW Helix | Greedy |
| CL4 | Shear + TI 6% | CL | CCW Helix | Greedy |

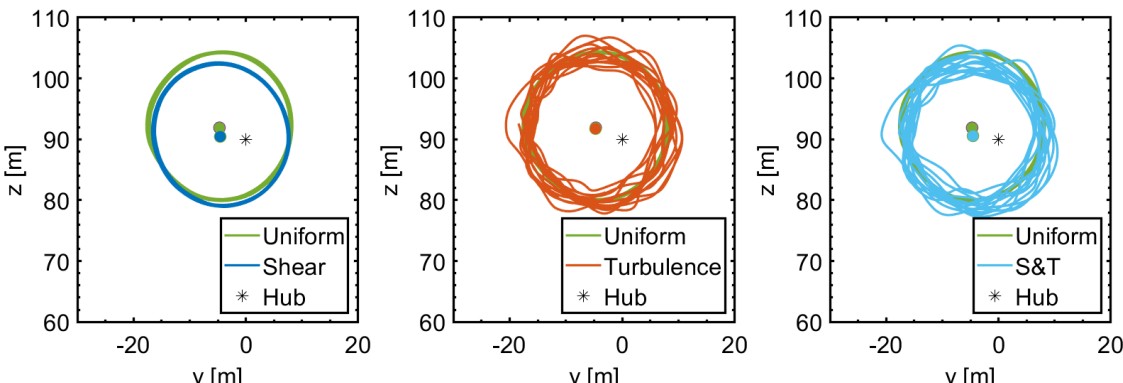

**Figure 10.** Comparison of hub vortex trajectory in the fixed frame of open-loop cases in uniform wind condition (OL$_1$) to shear (OL$_2$), turbulence (OL$_3$), and combined shear and turbulence (OL$_4$). "Hub" denotes the hub of the wind turbine.

(2) **Turbulence** Compared to the Helix in the uniform wind case, OL$_2$ shows increments in power production and fatigue for both turbines. This can be explained by the increased TI and the natural mixing effects of turbulence (van den Berg et al., 2023).

(3) **Combined Shear and Turbulence** When both shear and turbulence are present, the wind farm has a cumulative loss in power production and cumulative increments in both flapwise and edgewise DEL.

The comparison offers insights into the expected behavior of the closed-loop controller. However, it is crucial to acknowledge the difficulty in defining a definitive reference point as priorities vary across different interests. In this study, the main goal is to eliminate variations in the helical wake introduced by external wind conditions, generating a more consistent helical wake





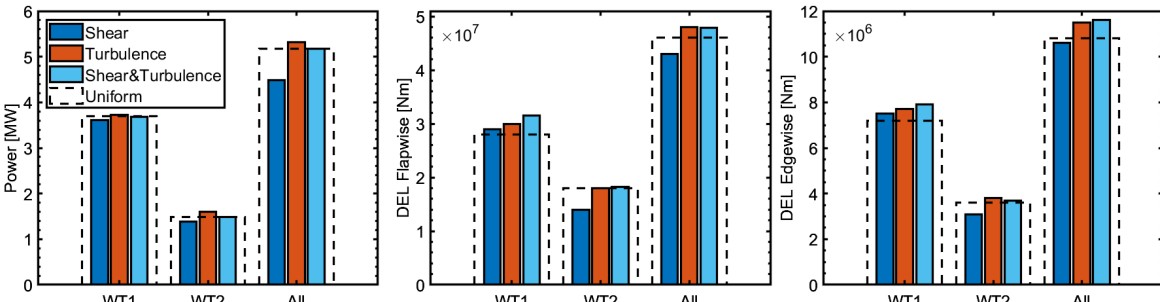

**Figure 11.** The power and DEL of $OL_2$ (blue), $OL_3$ (orange), and $OL_4$ (sky blue) compared to case $OL_1$ (dashed line). "WT1", "WT2", and "All" denote the upstream turbine, the downstream turbine, and the entire two-turbine wind farm.

relative to the uniform wind case. Thus, the output of $OL_1$ in the Helix frame is used as the reference for all cases. The corresponding objectives of the controller in terms of the hub vortex trajectory can be summarized:

(1) **Uniform** The performance of $CL_1$ should match that of the $OL_1$ by generating an identical Helix.

(2) **Shear** Compared to $OL_2$, the controller should rectify the steady-state bias.

(3) **Turbulence** The controller is expected to mitigate the extra oscillation of the hub vortex introduced by the turbulence. However, because the dominant turbulence frequency exceeds the roll-off frequency of the controller, this mitigation effect is going to be compromised. Thus, complete stabilization of the hub vortex rotation is unlikely. Nevertheless, the controller will still attempt the mitigation by generating dynamic pitch inputs.

(4) **Combined Shear and Turbulence** The controller should correct the bias while trying to mitigate the oscillation as much as possible.

### 4.3 Closed-Loop Framework Performance

This chapter shows the performance of the closed-loop system. In uniform wind conditions, both the open-loop and closed-loop systems generate identical helical wake structures, confirming the effectiveness of the proposed framework. For brevity, only cases where shear and turbulence are added are presented.

Figure 12 illustrates the hub vortex trajectory generated by the closed-loop system compared to the reference. The corresponding change in the wind farm performance compared to the open-loop is shown in Fig. 13. Based on the results, the following observations can be drawn:

(1) **Shear** The closed-loop system effectively corrects the steady-state bias, enhancing the power output of the downstream turbine. However, this improvement is accompanied by increased DELs for both turbines.





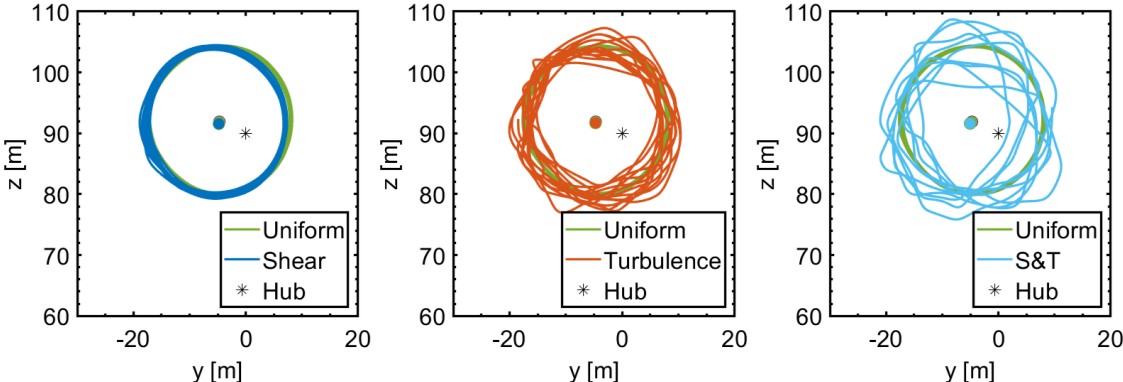

**Figure 12.** Comparison of hub vortex trajectory in the fixed frame of the uniform wind open-loop case ($OL_1$) to closed-loop cases under shear ($CL_2$), turbulence ($CL_3$), and combined shear and turbulence ($CL_4$).

(2) **Turbulence** The closed-loop system behaves as expected: the additional oscillation is not fully stabilized, so the power output of both turbines remains largely unchanged. However, the blade DEL for the downstream turbine rises, indicating higher fatigue loads.

(3) **Combined Shear and Turbulence** The closed-loop system corrects the shear-induced steady-state bias without mitigating the extra oscillations as expected. This results in increasing power outputs for the downstream turbine and the overall wind farm, at the expense of higher fatigue across both turbines.

The above observations demonstrate the closed-loop system's capability to address steady-state bias while revealing its limitations in handling disturbances, motivating further analysis of the underlying mechanisms.

## 4.4 Result Analysis

This section analyzes the obtained results for different wind cases, aiming to address the underlying reasons for the observed system behavior and assess its consistency with existing findings and literature.

### 4.4.1 Shear

In the case of shear, an increase in power production for the downstream turbine is noticed, due to the correction of the bias. This redirection of the helical wake results in a higher average inflow wind speed for the closed-loop case ($CL_2$) at 4D downwind position compared to the open-loop case ($OL_2$). The increased DEL and reduced power of the upstream turbine result from more aggressive control actions required for bias correction. Compared to the constant pitch inputs of $\beta_{\text{tilt}}^e$ and $\beta_{\text{yaw}}^e$ (Helix frame) in case $OL_2$, the closed-loop controller superimposes additional oscillatory components on both $\beta_{\text{tilt}}^e$ and $\beta_{\text{yaw}}^e$ as Fig. 14 shows, which likely contributes to increased fatigue of the upstream turbine.



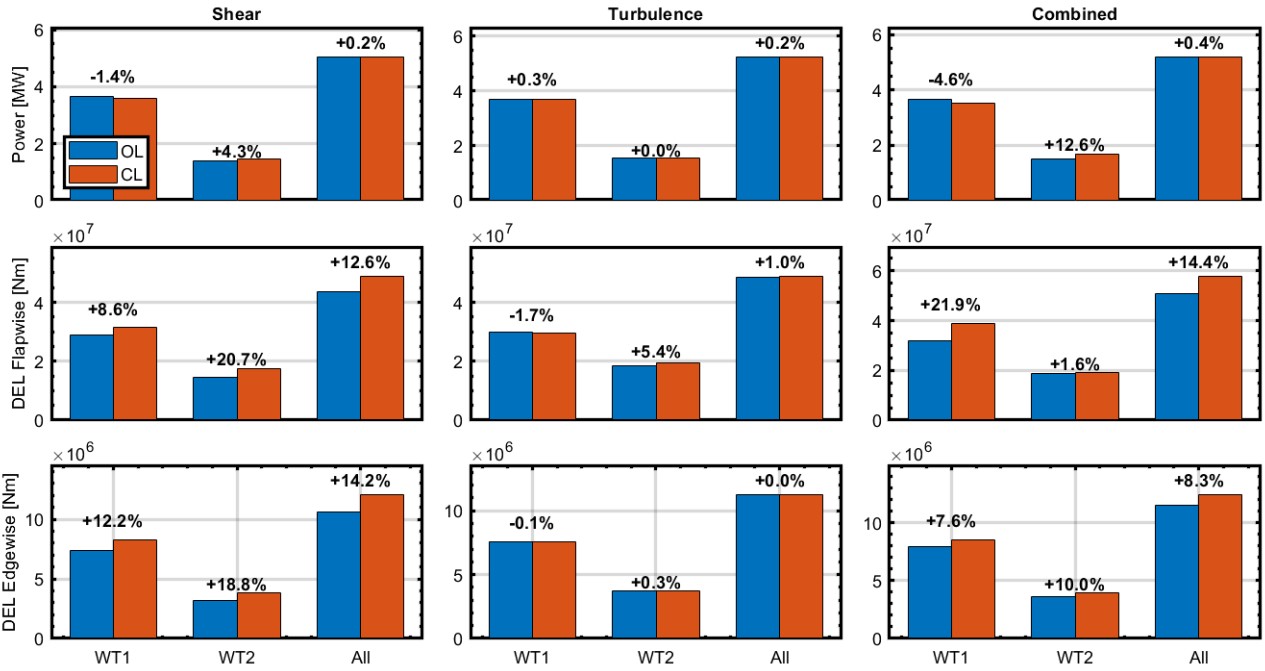

**Figure 13.** Comparison of wind farm performance between open-loop (blue) and closed-loop (orange) systems under three wind conditions.

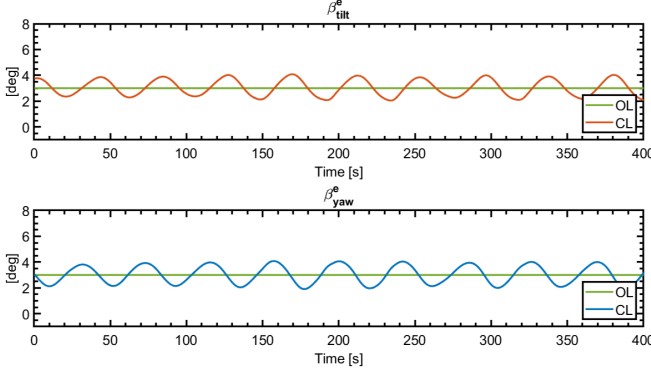

**Figure 14.** The comparison of pitch inputs in the Helix frame between the open-loop ($OL_2$) and closed-loop ($CL_2$) case under shear. Compared to $OL_2$, the additional oscillating components in $CL_2$ of both channels are obvious.

### 4.4.2 Turbulence

As analysed, the comparison of the hub vortex trajectory between the open-loop ($OL_3$) and closed-loop case ($CL_3$) indicates that there is no obvious improvement in the hub vortex movement, because the dominant turbulence frequency exceeds the





controller's roll-off frequency. Nevertheless, Fig. 13 indicates a slight increase in the power production, driven mainly by the
405 upstream turbine. To interpret this, the pitch input signal between case OL$_3$ and CL$_3$ is compared.

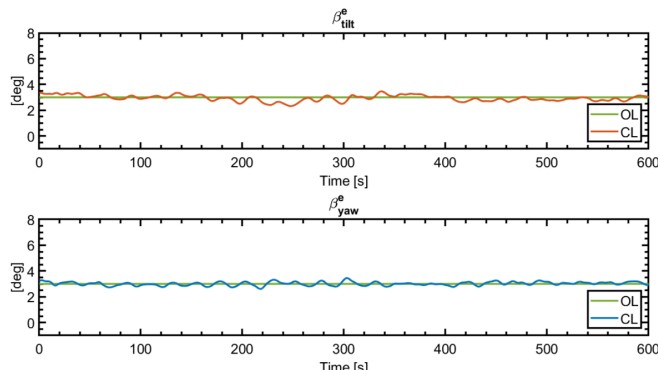

**Figure 15.** The comparison of pitch inputs in the Helix frame between the open-loop (OL$_3$) and closed-loop (CL$_3$) case under turbulence. Compared to OL$_3$, the pitch inputs of CL$_3$ are more dynamic.

Figure 15 presents the comparison: Informed by wake measurements, the closed-loop controller generates more dynamic pitch inputs compared to the open-loop counterpart while achieving a lower time-averaged magnitude. This explains the power increment in the upstream wind turbine, as proved by the work of Taschner et al. (2023), whereas the downstream power remains largely unchanged due to dominant natural turbulent mixing. This shows that incorporating wake measurement enables
the controller to reduce unnecessary actuation relative to a fixed input, and thereby improves overall wind-farm performance. Thus, further supports the value of incorporating wake measurements into the control loop.

Finally, flapwise DEL increases for the downstream turbine and the farm overall, while edgewise DEL remains nearly unchanged. This aligns with van Vondelen et al. (2023), which reports the edgewise DEL's lower sensitivity to variations in Helix magnitude.

### 4.4.3 Combined Shear and Turbulence

In this case, the helical wake bias is corrected, but the oscillations remain, as discussed previously. This leads to an increase in power production for the wind farm at the cost of higher fatigue and power loss in the upstream turbine due to stronger Helix actuation. Compared to CL$_2$, the power gain is larger due to stronger mixing caused by the inherent natural mixing in the turbulence.

## 5 Conclusions

This study proposed a framework for closed-loop wake mixing control, with a focus on the Helix approach. A downwind-facing continuous-wave LiDAR is used to sample the hub vortex as the controlled variable, and a control system is designed to track the target hub vortex position in the Helix frame. Simulations show that the framework performs as intended in uniform




wind conditions and shows effectiveness for the downstream turbine when shear is involved. In the latter case, the controller
successfully compensates for the steady-state bias in the hub vortex trajectory, resulting in a power increase of $4.3\%$ and $12.6\%$
for the downstream turbine under shear and combined shear-turbulence conditions. Performance is limited under turbulence
due to a controller roll-off induced by non-minimum phase zeros. Nevertheless, wake measurement-informed pitch adjustments
yield modest upstream and farm-level power gains with minimal load increases, reinforcing the value of incorporating wake
measurement for closed-loop control. Thus, the framework offers a novel flow-informed strategy for wake mixing control.
Future work will explore LES validation, realistic LiDAR integration, improved reference design, and controlled variable
selection by incorporating better aerodynamic information of the wake.

*Code and data availability.* A free-to-use version of QBlade can be found at https://qblade.org (Marten et al., 2013). The software used for
postprocessing the data can be found at the 4TU repository at https://doi.org/10.4121/22134710.v2. At that repository, two README files
explain how the MATLAB scripts can be used.

*Author contributions.* ZC led the conceptualization, methodology, software development, validation, formal analysis, investigation, visual-
ization, and original draft preparation. AAWvV contributed to conceptualization, manuscript review, supervision, and editing. JWvV con-
tributed to conceptualization, supervision, manuscript review, and editing. All authors provided feedback on the methodology and reviewed
the final manuscript.

*Competing interests.* At least one of the (co-)authors is a member of the editorial board of Wind Energy Science.

*Acknowledgements.* The author gratefully acknowledges the support of the Data-Driven Control group at the Delft Center for Systems and
Control (DCSC), TU Delft. We also extend our thanks to Dr. Daniel van den Berg for his assistance with QBlade.



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
