# Peer review of "LiDAR-enhanced Closed-Loop Active Helix Approach"

_Wind Energy Science, 2025_

## Author Comment (AC1)

| Date | January 8, 2026 |
|---|---|
| Our reference | wes-2025-161 |
| Your reference | n/a |
| Contact person | Zekai Chen |
| Telephone/fax | +31 (0)64 87 13031 / n/a |
| E-mail | chenzk0429@gmail.com |
| Subject | Response to Reviewers |

**Delft University of Technology**

Delft Center for Systems and Control

Address
Mekelweg 2 (3ME building)
2628 CD Delft
The Netherlands

www.dcsc.tudelft.nl

Anonymous Reviewer #1
Anonymous Reviewer #2
*Reviewers, Wind Energy Science*

Dear Reviewers,

We sincerely thank you for the constructive and thoughtful feedback on our manuscript *LiDAR-enhanced Closed-Loop Active Helix Approach* (Ms. Ref. No.: wes-2025-161). We have revised the paper thoroughly. Below, we respond point by point, quoting each comment and indicating the changes made. A clean manuscript with highlighted changes is provided in the resubmission.

Yours sincerely,

Zekai Chen
Aemilius A.W. van Vondelen
Jan-Willem van Wingerden

Enclosure(s): Response to comments of Anonymous Reviewer #1
Response to comments of Henrik Asmuth
Response to comments of Anonymous Reviewer #2

**Response to comments of Anonymous Reviewer #1**

**Reviewer:** *In the manuscript "LiDAR-enhanced Closed-Loop Active Helix Approach", the authors demonstrate a control algorithm which uses backward facing lidar to identify the hub vortex motions, and use this information to control the helical wake. This is a very promising line of study, as previous works have only considered the open-loop formulation, and the addition of lidar information opens the door towards many possibilities for optimizing active wake mixing approaches.*

**Authors:** We thank the reviewer for the positive comments and appreciate the time and effort to review our work.

**Reviewer:** *There are two major items to consider in this manuscript which might strengthen the study and bolster the arguments presented here. First, in section 4.2, it mentions that the main goal of the closed loop algorithm is to "eliminate variations in the helical wake introduced by external wind conditions, generating a more consistent helical wake relative to the uniform wind case." While there are many possible objectives for a closed loop controller, it is unclear if this leads to the best possible outcome in a wind farm context. There may also be external conditions in which this objective cannot be reached, such as in cases with atmospheric stratification or cases with high veer, or where the cost of matching the uniform wind case would actually be detrimental to the upstream/downstream turbine operation (this is alluded to in section 4.4.3 with combined shear and turbulence). The authors could consider what might be alternative objectives to apply, and also discuss the choice of objective more prominently in earlier sections of the manuscript.*

**Authors:** We thank the reviewer for highlighting this important point. The objective of the closed-loop controller can indeed go beyond simply replicating the open-loop behavior under the same wind conditions. In this study, our goal is to provide a proof-of-concept demonstration, illustrating the possibility of incorporating downwind-facing LiDAR measurements into a closed-loop control framework for wake mixing. Thus, we did not explore more options for the reference given. Nevertheless, we acknowledge that this approach allows for a broader range of possible control objectives in practical implementations. In the latest manuscript, we have added more discussion about the closed-loop objective in section 4.1, where the simulation setup is introduced.

**Inserted text:** In this study, we chose an initial goal to eliminate variations in the helical wake introduced by external wind conditions, generating a more consistent helical wake relative to the uniform wind case. This goal is set by assuming that the wind farm stakeholders value the consistency in wind farm performance to ensure predictability of power production and reduce operational risk [1]. In our simulation, we found that the performance of the wind farm varies under different wind conditions, accompanied by changes in helical wake structure. And thus, we aim to ensure this consistency by eliminating the variation in the helical wake. As a result, the output of $OL_1$ in the Helix frame is used as the reference for all cases. However, this assumption does not guarantee the optimality of the decision. Additionally, it is crucial to acknowledge the difficulty in defining a definitive reference point as priorities vary across different interests, and thus, the given reference varies. For instance, as shown in a later chapter, both the upstream and downstream turbines have significantly increased load under shear despite increases in power. Thus, eliminating variations in the wake might not be a good idea in this case; instead, a reference that balances the power production increase and the fatigue load increase could be given. Furthermore, in the case of extreme wind conditions like high veer, the chosen reference may simply be infeasible. Therefore, the reference signal should be selected flexibly depending on the operator, and future studies should explore more condition-based reference choices.

**Reviewer:** *Secondly, the current study focuses exclusively on applying the control algorithm on variations of one inflow wind speed at 10 m/s. The study also assumes a priori knowledge of the inflow wind speed, as it does not appear to be derived from other control inputs to the turbine. While this simplifies the initial demonstration of the approach, it also introduces several questions. It is currently unclear how much modification of the algorithm is required to handle the more complex problem where the wind speed changes (see comment 9 below), and whether the results of section 4 are similar at other wind speeds. At wind speeds past rated, the turbine blade pitch changes as a function of the wind speed, so it could be worth considering whether the closed loop algorithm can robustly handle those changes.*

**Authors:** We thank the reviewer for raising this point. We have updated the latest manuscript to include the discussion of this point in the "Discussion" chatper.

Our study indeed assumes a prior knowledge of a uniform constant inflow wind speed of 10 m/s. However, as the reviewer correctly noted, varying average inflow speed would result in several modifications to the proposed framework:

First, the input tilt ($\beta_{\text{tilt}}$) and yaw ($\beta_{\text{yaw}}$) pitch signals will change accordingly as the Strouhal number is kept at the constant optimal value of 0.25 [12], as the definition of St shows:

$$\text{St} = \frac{f_e D}{U_\infty},\tag{1}$$

This change of actuation frequency will subsequently influence the fatigue load of the individual wind turbine [26]. Subsequently, the time delay $\tau$ that the Smith predictor utilizes will require corresponding updates since this value changes with the mean inflow wind speed $U_\infty$. These modifications highlight the need to include an inflow wind speed estimator and comprehensive testing under time-varying wind conditions in future works.

In practice, the wind field entering a wind farm varies continuously [7]. Hence, an accurate estimate of the free mean wind speed is important for the performance of the closed-loop controller [14]. Current literatures offer several ways of achieving this. One possible approach is presented in [3], in which the inflow wind condition is estimated based on the blade load measurements and through the classical blade element momentum (BEM) theory. This is similar to the technique adopted in the study of [20], where local blade inflow measurements, including the angle of attack and the relative velocity, are used for load mitigation. An alternative method is offered in [9], where an Ensemble Kalman Filter (EKF) incorporates turbine measurements into a real-time state estimator for closed-loop wind-farm control. The built observer achieves a great balance between computational cost and estimation accuracy. A further option is shown in the work of [21], where an Immersion and Invariance Wind Speed Estimator is developed for wind estimation.

Lastly, we understand the reviewer's concern about the functioning of the proposed method above the rated wind speed. This situation is not considered in this work since the Helix approach is typically implemented at below-rated wind conditions, where power losses are more substantial compared to those encountered in above-rated conditions [11].

**Inserted text:** This study assumes a prior knowledge of a uniform constant inflow wind speed of $10 \text{m/s}$, however, the wind field entering a wind farm varies continuously due to natural processes [7]. Hence, an accurate estimate of the free mean wind speed is important for the performance of the closed-loop controller [14]. One possible approach is presented in [3], in which the inflow wind condition is estimated based on the blade load measurements and through the classical blade element momentum (BEM) theory, similar to the work of [20]. An alternative method is offered in [9], where an Ensemble Kalman Filter (EKF) incorporates turbine measurements into a real-time state estimator for closed-loop wind-farm control, achieving a great balance between computational cost and estimation accuracy. A further option is shown in [21], where an Immersion and Invariance Wind Speed Estimator is developed. As a result, for future works, incorporating these methods into the proposed framework for inflow wind speed estimation enables a more adaptive system.

**Reviewer:** *The current work assumes that the dynamics of the hub vortex is properly resolved in QBlade. However, modeling the dynamics of the wake immediately aft of the hub and nacelle (<1D downstream of the rotor) can be challenging, even for actuator line model or blade-resolved simulations. Are there parameters which control the hub and nacelle properties, such as drag coefficient or nacelle area, in QBlade which might impact the hub vortex dynamics? Providing some additional information on the hub and nacelle model used in QBlade could be helpful for readers.*

**Authors:** We appreciate the reviewer for the suggestion. In the latest manuscript, the parameters of the hub and nacelle properties are given in section 2.3, where we introduced the NREL5MW turbine and QBlade.

**Inserted text:** This work assumes that the dynamics of the hub vortex aft of the hub and nacelle are adequately resolved. However, this can be challenging in practice. In Qblade, the hub and nacelle properties of the NREL5MW wind turbine, including mass and inertia, are specified in Table 1.

**Reviewer:** *In section 3.2.1, the lidar sampling plane is given as encompassing the full rotor plane at a distance of 1D downstream from the rotor. Some additional details on the numerical lidar sampling should be provided, including the sampling frequency, the total number of points sampled, and the spatial distribution of the sampling points.*

**Authors:** We appreciate the reviewer drawing attention to this oversight. Some additional information has been inserted where the LiDAR model is introduced.
**Inserted text:** Furthermore, the LiDAR captures the flow information across the entire rotor disk by simultaneously sampling 80 points uniformly distributed in the Cartesian coordinate system. The number of sampling points is determined to balance the spatial resolution and the computational cost. Lastly, the LiDAR is modeled to have a sampling frequency of $10\,\mathrm{Hz}$ to ensure consistency with the QBlade simulation time step of $0.1\,\mathrm{s}$.

**Reviewer:** *Related to the point above, there may be some practical considerations when using backward-facing lidars to measure turbine wakes. Recent field measurements of turbine wakes using a continuous wave spinner lidar require designing specific scan patterns (e.g., the rosette scan pattern in Hsieh et al, $https://doi.org/10.1016/j.jweia.2021.104754$) with restrictions on revisit times and probe volumes. For a turbine the size of the NREL5MW, the revisit times and the minimum probe volumes may be fairly large (on the order of 10s of meters), and that may reduce the resolution of the measured wake. The authors may wish to comment on some of these practical aspects as they relate to the current study, e.g., would the averaging effect of the lidar sampling impact the helix feature extraction?*

**Authors:** We thank the reviewer for raising this point. It is true that there are several things to be considered when implementing LiDAR for Helix feature extraction in practice. In the revised version of the manuscript, we discussed the practical usage of LiDAR more promptly in the newly added "Discussion" part of the paper.

The discussion unfolded based on these sources of errors: time and spatial averaging effect, probe measurement volume, and LiDAR instrument error. The work of [16] examines the effect of errors, introduced by the time and the spatial averaging effect, on the wake measurement. This work is done by comparing the horizontal and vertical wind profile measurements between a simulated LiDAR and a real LiDAR, focusing on mean velocity deficits and turbulence intensity. Overall, the LiDAR successfully captured the wake profile trends observed in the experimental data [16]. The study shows that the simulated LiDAR corresponds well with the mean velocity deficit in the horizontal profile, especially near the center of the wake, but significantly underpredicts the peak turbulence intensity at the edge of the wake. This attenuation of the turbulence content arises from spatial averaging along the probe line-of-sight, a known shortcoming of LiDAR measurements [4]. For the vertical profile, the simulated LiDAR significantly underpredicts the wake deficit near the center of the wake, although the deficit at the wake edges and the corresponding turbulence intensity are captured adequately. Since the proposed framework relies on the velocity difference between the hub vortex and the surrounding wake to extract the Helix feature, these discrepancies in wake deficit should be addressed. Therefore, the filter threshold of the hub vortex needs to be retuned when implementing the proposed framework in practice.

The study of [24] further examines the influence of range weighting errors and the directional bias error, stemming from the weight function and the line-of-sight measurement angle, on a feedforward LiDAR's wake measurement. Results show that range-weighting errors increase with focal distance, while directional bias increases with the scanning angle. Although this work focuses on the feedforward LiDAR, it offers some insights into using a backward-facing LiDAR. In the proposed framework, we assumed full information acquisition, 1 rotor diameter behind downwind of the wind turbine. This, in reality, however, might lead to decreased resolution of the wake measurements due to the range weighting errors and the directional bias errors. Thus, it is important to choose appropriate parameters of the LiDAR when implementing it in reality.

Lastly, both [24] and [6] analyze the influence of instrument-related errors on LiDAR-based wake measurements. According to [6], the continuous-wave LiDAR generates a mean error between $-0.2$ and $0.2$ m/s and aleatoric uncertainties with a standard deviation of $0.20$ m/s in measuring the wake compared to a cup anemometer at the same height. Such errors affect the choice of threshold used in the Helix feature extraction filter. Therefore, a calibration based on the field measurement is recommended when applying this framework in practice.

**Inserted text:** The LiDAR adopted in this study assumed perfect wake acquisition 1 rotor diameter downstream of the upstream turbine. However, real LiDAR systems may violate those assumptions, introducing various sources of uncertainty to the wake measurement. The work of [16] compares the wake measurement of a simulated LiDAR and a real LiDAR. Overall, the study finds that the simulated LiDAR is able to capture the trend of wake successfully. However, the accuracy of LiDAR's measurement in the wake deficit and turbulence intensity decreases near the center of the wake due to the time and spatial averaging effect. Additionally, the work of [24] demonstrates that the weight function and line-of-sight measurement introduce range weighting errors and the directional bias error, further reducing the wake measurement accuracy. Finally, the instrument-related errors of a realistic LiDAR, such as height-resolution limitations and environmental influences, compromise the wake measurement, as [6] denotes. As a result, the proposed pipeline needs to be retuned when applying the proposed framework with a realistic LiDAR. It is highly recommended to perform a field calibration before adopting the proposed framework in practice.

**Reviewer:** *In section 3.3, a delay time of T=11.2 seconds was determined. Assuming that a lidar samples at a distance of 1 rotor diameter for the NREL5MW (126m), this leads to a convection velocity of 11.25 m/s between the rotor plane and the lidar detection plane, which is faster than the mean inflow velocity of 10 m/s. In typical wake cases, we would expect the convection velocity of wake structures to be about 60-70% of the inflow velocity. Some details on the calibration process would be useful in this regard, as the value of alpha reported here seems to be in conflict with Taylor's hypothesis. Additionally, how sensitive is the control system to the value of alpha?*

**Authors:** We thank the reviewer for raising this point. This happens because the modeling of the nacelle in QBlade has very limited fidelity. As a result, consistent with the finding of [23], the generated hub jet has a higher wind speed compared to the average inflow wind speed. The work of [23] studied the effect of the nacelle and the tower on the wind flow passing through a wind turbine experimentally. As illustrated in Figure 1, a jet is generated at the height of the hub when the nacelle is not properly modelled. The corresponding contour plot shows that the velocity of the jet satisfies $\bar{U}/U_\infty > 1$ (estimated to be 1.1), meaning that the speed of the wake in the jet $\bar{U}$ is higher than the average inflow $U_\infty$. In our study, the dead-time delay $\tau$ is found to be 11.8 seconds, which implies that the $\bar{U}/U_\infty = (126/11.8)/10 \approx 1.08 > 1$, consistent with the finding of [23].

[Figure]

Figure 1: Contour lines of $\bar{U}/U_\infty$ in a vertical section, with $\bar{U}$ denoting the wind speed and $U_\infty$ denoting the average inflow wind speed. The top (A) represents the case without a tower and a nacelle, and the bottom (B) represents the one with a tower and a nacelle. The two vertical lines denote $1D$ and $3D$ downwind position with $D$ standing for the rotor diameter. In the contour line plot, the solid line represents the large eddy simulations result, and the dot represents the experiments. Figure adopted from [23].

The poor modeling of the nacelle does not influence the torque or power production of the wind turbine, and the distribution of the mean wind speed in the far wake, as studies [10] and [23] show. Thus, the author believes that the result of the variations in power production and fatigue load in this work still holds. However, both studies show that the wake dynamics will be impacted as the hub jet stabilizes the wake and delays the vortex breakdown [23]. Therefore, the dynamics and the accessibility of the hub vortex will be influenced. The work of [5] studies the vortex structures generated by the Helix approach in the large eddy simulation (LES), and found that the hub vorticity tends to rotate in the CCW direction when the CCW helix is activated in the near wake region of a wind turbine. This finding indicates that the data processing pipeline has a huge likelihood of still being able to hold as it is designed according to the moving pattern of the hub vortex. Therefore, future works should validate the hub vortex acquisition in a high-fidelity simulation environment.

Given the document above, we acknowledge the limitations of this work associated with the fidelity limitation of the QBlade platform. Future studies should validate the proposed framework in a high-fidelity simulation environment, such as the large eddy (LES) simulations, in which tower and nacelle aerodynamics are more accurately represented. The revised manuscript has discussed this limitation promptly. **Inserted text:** In QBlade, the nacelle is modelled with limited fidelity. While this is unlikely to influence the mean downwind velocity field in the far wake, it can influence the turbine power fluctuations and turbulence kinetic energy, as noted in the work of [10]. Moreover, [23] supports this experimentally. The study of [5] further shows that the rotating pattern of the hub vortex still holds in the LES simulation when the Helix approach is applied. This suggests that the data-processing pipeline developed in this work, which is based on tracking the motion of the hub vortex, remains applicable despite these modelling limitations. Nevertheless, future work should validate the proposed control framework using a high-fidelity simulation environment where the aerodynamics of the nacelle are more accurately modeled, enabling a more comprehensive assessment of the system's behavior.

**Reviewer:** *In line 297, a missing reference is present (it appears as a question mark "?").*

**Authors:** We thank the reviewer for noticing this. We have fixed this issue by correctly citing the reference.

**Reviewer:** *Section 4.1 mentions that a second turbine is placed 4 rotor diameters from the upstream turbine, due to limits of the simulation quality within QBlade. That streamwise spacing is relatively close, even for offshore turbines, and may miss some dynamics of the helix modes in the farther downstream (see G. Yalla et al, `https://wes.copernicus.org/preprints/wes-2025-14/`). This study could be strengthened by commenting on how this lidar methodology could apply, or the results might change, with different turbine spacings.*

**Authors:** We thank the author for raising this point. In the revised version of the manuscript, we discussed this promptly in the ending part of the paper.

The work of [28] demonstrated the frequency study of the wake after adopting different active wake mixing control methods, offering insights in answering the question of how the wind farm performance will change with different turbine spacing. Among the frequency content, the turbulence kinetic energy (TKE) is the main factor that determines the wake recovery effect, which subsequently influences the performance of the wind farm in terms of power production and fatigue load [28]. Additionally, different components in the wake are quantified and separated by azimuthal wavenumbers $\kappa_\theta$ and excitation frequency $\omega_\theta$, describing the shape and frequency of the flow structures that are imparted on the wake.

Figure 2 shows the contribution to TKE, the factor that quantitatively reflects the strength of wake mixing, of different components in the wake with increasing distance downwind. The larger the value, the more the component contributes. Based on the figure, the component $\kappa_\theta = -1$ (counterclockwise component generated by the Helix approach) contributes predominantly in the region between 0 and 1.5D approximately. However, as the distance increases, the contribution of this component decreases, and contributions of others, especially $\kappa_\theta = 0$ and $\kappa_\theta = -2$, increase. As the distance increases to 4D, the $\kappa_\theta = 1$ component replaces the counterclockwise component in contributing the most to the TKE, and subsequently the wake mixing. Finally, as distance increases, the contribution of $\kappa_\theta = \pm 1$ and $\kappa_\theta = 0$ rises with increasing sum-up value, indicating more wake recovery. This is consistent with the finding that wake deficit recovers naturally with increasing distance [25].

In our work, the downstream turbine is placed at 4D position, where components $\kappa_\theta = \pm 1$ and $\kappa_\theta = 0$ contribute dominantly. Based on the findings of [28], it can be expected that the wake recovery will improve with larger turbine spacing. However, determining how the proposed framework performs under increased spacing requires a better understanding of how the controller modifies the flow structures. To the authors' knowledge, such an analysis has not yet been done. Consequently, we recommend that future research investigate the interaction between the proposed framework and wake dynamics across varying turbine spacings, both qualitatively and quantitatively.

[Figure]

Figure 2: Model contribution to TKE of different components in the wake, quantified by the normalized eigenvalues $|\lambda_i|$ from the spectral proper orthogonal decomposition eigenvectors at $\mathrm{St} = 0.3$ and $\kappa_\theta = 0, \pm 1, \text{and} \pm 2$. Figure adopted from [28].

**Inserted text:** To further improve the overall performance and fully utilize the potential of the proposed framework, future works should consider conducting better feature extraction and performing quantitative flow analysis, similar to the study of [28], to directly facilitate wake mixing and better understand the influence of the proposed framework on wind flow.

**Reviewer:** *In table 2, a jet height of 100m is mentioned. Is a low-level jet wind profile being considered in this study?*

**Authors:** We thank the reviewer for pointing this out. In this study, we used the power law wind profile instead of a lower-level wind profile, also specified in Table 2. In QBlade, this value is only valid when the wind profile is selected as "JET" [17], which is not the case here. In the revised version of the manuscript, the author has deleted this entry in the form to cause less confusion. We thank the reviewer again for pointing this out.

**Reviewer:** *Some details on the nominal turbine controls for the NREL5MW, outside of the helix and closed-loop control method discussed in this paper, should be included in section 4.1 or 4.2. For instance, are the blade pitch and rotor speed settings constant, determined by pitch or rotor speed schedules, or set by external control algorithms (e.g., ROSCO https://github.com/NREL/ROSCO) ?*

**Authors:** We appreciate the reviewer for pointing out this potential improvement. We have added the following text in section 4.1 to introduce details of the nominal turbine control system for readers.
**Inserted text:** The NREL5MW turbine is selected as the object of study. Its power-production operation is governed by two independent control systems: a generator-torque controller and a collective blade-pitch controller, designed to operate in the below-rated and above-rated wind speed range, respectively [18]. The goal of the former is to maximize power capture below the rated wind speed, whereas the latter aims to regulate generator speed above the rated wind speed. In reality, the Helix approach is typically implemented at below-rated wind conditions, where power losses are more substantial compared to those encountered in above-rated conditions [11]. Consequently, a $k\omega^2$ torque controller is employed to maintain an optimal tip speed ratio to maximize the power production. The gain constant $k$ is set to 2.3323 based on [18].

**Reviewer:** *When applied to more realistic scenarios, the inflow wind speed is not constant, which means that the excitation frequency $f_e$ is not constant. How would this situation be handled or incorporated into the current closed-loop algorithm, as there may be additional averaging/time delays involved in determining the appropriate excitation frequency and adjusting the closed-loop algorithm?*

**Authors:** We thank the reviewer for pointing this out. This question has been discussed in the early part of the cover letter.

**Response to comments of Henrik Asmuth**

**Reviewer:** *The Helix-visualization in Fig. 2 seems to be from Korb et al (2023), `https: // doi. org/ 10. 1017/ jfm. 2023. 390`. Please cite the paper if you use the image.*

**Authors:** We thank the reviewer for noticing this oversight. We have appropriately cited the origin of the figure in the caption of Figure 2.
**Inserted text:** The helical wake figure is adopted from the work of [19].

**Response to comments of Anonymous Reviewer #2**

**Reviewer:** *Overall, I find this paper to be a very interesting exploration of helical wake improvements using remote sensing measurements. The analysis given is rigorous, with a LiDAR data processing pipeline implemented, and combination of numerous advanced wind turbine control developments such as a helix frame transformation for analyzing the hub position and H-infinity control synthesis.*

**Authors:** We thank the reviewer for the positive comments and appreciate the time and effort to review our work.

**Reviewer:** *The analysis provided in this paper is strong. However, there are a few points that could be elaborated/explored further. When analyzing the data in Figure 13, power increases in the downstream wind turbine are highlighted; the "All" section of the bar plot is not mentioned. When looking at this column, there is a 0.4% power increase overall, with an 8-14% increase in blade loads. These load increases relative to the overall power increase are very concerning and should be addressed in some way to not leave the reader wondering. In addition, no error bars are provided, raising additional concerns since an error margin of $> 0.4\%$ could be somewhat reasonable.*

**Authors:** We thank the reviewer for pointing this out. In the revised manuscript, we discussed the power-fatigue trade-off more promptly in the "Discussion" chapter. We acknowledge the reviewer's concern regarding the relatively small magnitude of the increases in the shear situation, which might give the impression that the variations stem from numerical randomness. First, some small magnitude increments correspond to the entire two-turbine wind farm, whereas the magnitude of changes corresponding to the individual turbine responses is physically reasonable. Secondly, the goal of this study is to demonstrate, as a proof of concept, that incorporating flow measurements into closed-loop wake-mixing control is feasible, rather than to specifically optimize power production or minimize loads. For this reason, simulations were performed using a single wind seed to show this feasibility. To strengthen the analysis in future work, simulations with multiple wind seeds should be conducted. We have mentioned this point in the "Future Work" session in the revised manuscript. Furthermore, in the revised version of the manuscript, we have mentioned this point in the "Conclusion" section.
**Inserted text:** Moreover, the results presented in this paper are obtained from a single simulation with one wind seed, as this study primarily aims to serve as a proof-of-concept to demonstrate the feasibility of incorporating flow information into dynamic wake mixing control. Future work should therefore consider simulations with multiple wind seeds to mitigate the influence of numerical randomness and to provide a more comprehensive characterization of wind farm behavior.

**Reviewer:** *Overall, there is no inherent problem in Figure 13; rather, there should be more text elaborating the data that is present. This could be done through more body text in the results section, or through more descriptive captions. Throughout the paper, the captions for figures are sparse, leaving a lot of abbreviations and annotations ambiguous. For example, Table 3 does not explicitly define what "WTi" or "Greedy" means. The meaning of these terms can be assumed based on context, but adding an additional sentence to the caption could help disambiguate these terms.*

**Authors:** We appreciate the author for pointing this out. In the latest version of the revised manuscript, the author has added additional captions to clarify the abbreviations.

**Reviewer:** *The idea of using closed-loop control to improve hub wake positioning in shear and turbulent conditions is compelling, but its actual success in industry seems somewhat dubious when comparing the increase in blade loads with the increase in power production. While it would be outside the scope of this paper to do a complete techno-economic analysis on the net benefit despite increased loads, this downside should be discussed in more detail. Even open-loop helical wake generation increases blade loads due to individual blade pitching at each revolution. Therefore, a modest discussion of why closed-loop helical wake generation should be used with or in place of other methods, such as wake steering, would increase the impact of this paper. Such a discussion could be placed right before (5) Conclusion.*

**Authors:** We thank the reviewer for highlighting this limitation. A dedicated discussion chapter has been added before the conclusion chapter to discuss this trade-off more promptly.

**Inserted text:** Simulation results presented in Chapter 4 indicate that the proposed framework has the best performance under turbulence, where an increase in power production is accompanied by a minor increase in fatigue load. However, the significant increase in fatigue load when shear is present indicates that a balance needs to be found between increasing power production and fatigue load. Although such an increase in power may be attractive to operators during periods of high electricity prices, the accompanying growth in fatigue loads must be carefully considered. Under shear conditions, the increased fatigue load of the upstream turbine can be explained by the additional pitch movement generated to correct the shear-induced steady-state bias. However, the mechanism leading to the increased fatigue load for the downstream turbine remains unknown. Therefore, further study should be conducted to understand this behavior.

Lastly, these findings yield two insights. First, the proposed framework may be better deployed under turbulent wind conditions rather than shear-dominated conditions. Furthermore, our initial strategy of moving the helix center back to the uniform open-loop counterpart may not be optimal. To improve the performance of the proposed framework under shear, further studies should therefore be conducted to examine the effect of shifting the helix center to different positions and assess how these choices influence the wind farm's performance and the corresponding operation cost. These analyses would ultimately support a more balanced wind farm performance.

**Reviewer:** *The Methods used in this paper are very well presented and thorough. For example, the MBC transform and LiDAR modelling methods were very well established. However, there was a lack of discussion of the Internal Model Identification. It is not established clearly why the authors decided to use system identification instead of an analytical physical derivation, or even gray box modeling of unknown system parameters. A brief elaboration on why this design choice was made would be insightful for the reader.*

**Authors:** We appreciate the reviewer for pointing this out. The revised manuscript has added a more detailed explanation of the modeling choices to ensure clarity.

The primary reason for using system identification to create a linear model for control is to simplify the controller design. Specifically, the purpose of utilizing the Helix frame transform is to simplify the entire workflow by mapping both the input ($\beta_{\text{tilt}}^e$ and $\beta_{\text{yaw}}^e$) and output ($z$ and $y$) signals to the Helix frame, where the signals exhibit linear and static behavior. Thus, a self-identified linear model is selected to achieve this simplicity. Furthermore, adopting a linear model enables the real-time deployment of the proposed framework, thanks to its small computational cost compared to an analytical physical derivation model. Finally, because this study focuses on local wind-turbine control, large steady-state, control-oriented wind-farm models such as FLORIS [8] or FLORIDyn [2] are not employed. Nevertheless, if the framework were to be extended to farm-level applications, incorporating such models would be highly beneficial.

**Inserted text:** The primary motivation for deriving this model is to simplify the subsequent controller design and to ensure computational efficiency. Since the proposed framework is intended for real-time implementation, the surrogate model must maintain a low computational cost. Since both input and output signals are mapped to the Helix frame, where they exhibit approximately linear behavior, building a linear model is an effective choice.

**Reviewer:** *Also, there could be more discussion on how the nonminimum phase zero is limiting the system performance. It seems like the system bandwidth reduction was chosen due to work done in Skogestad and Postlethwaite (2005). However, it is not made clear where the nonminimum phase zero is coming from (e.g., from actuator limitations, or from the helix generation dynamics themselves). This is a very important point to establish, since the nonminimum phase zero is preventing any improvements in hub vortex tracking in turbulent conditions.*

**Authors:** We appreciate the reviewer's feedback. We have revised the manuscript to add more comments on the nonminimum phase zero. Overall, there are three primary sources that may give rise to the nonminimum phase zeros. The first originates from the actuator delay that is not fully compensated by the Azimuth offset $\psi_{\text{off}}$. As demonstrated in the work of [15], the large phase lag introduced by the pitch actuator reduces the performance of the Individual Pitch Controller (IPC) considerably. This delay manifests as a coupling between the tilt and yaw channels subsequently in the non-rotating frame, which can be shown directly from the definition of the MBC transform. Although this delay can be mitigated by including an azimuth offset in the inverse MBC transform [22], perfect decoupling was not achieved in this study, evident by the non-identical RGA matrix. Thus, the uncompensated delay contributes to the right-hand-plane (RHP) zeros, and subsequently, the nonminimum phase behavior of the system.

Another source that contributes to the RHP zeros is the sensing delay inherent to the downwind measurement mechanism. Since the proposed framework controls the generated wake by relying on wake measurements acquired downstream, a time delay is naturally introduced into the loop. Although a Smith predictor with a nominal model was used to compensate for this effect, unmodeled or mismatched delay components likely remain and contribute to the nonminimum-phase zeros.

Finally, we noticed an inverse transient response of the identified system in the step response, an indication of nonminimum phase behavior. However, it is not clear yet whether the intrinsic dynamics of the helical wake itself introduce nonminimum-phase behavior. The study of [5] represents the closest work toward answering this question, in which investigations were conducted to study the interaction between the wind turbine under the Helix approach and the generated vortex structures in both the near and far wake regions using LES. However, that study focuses primarily on time-averaged vortex behavior instead of the transient response of the system. As such, the existing literature does not resolve whether the wake dynamics contribute to the right-half-plane zeros. Future research should therefore investigate the transient helix dynamics to determine whether they play a role in the emergence of nonminimum-phase behavior.

**Inserted text:** These nonminimum phase zeros are likely introduced by unmodeled delays in the system: the residual pitch actuator delay that is not fully compensated by the Azimuth-offset $\psi_{\text{off}}$, as well as the inherent downwind sensing delay. It remains unclear whether the intrinsic dynamics of the helical wake also contribute to this nonminimum-phase behavior. Therefore, future work should investigate whether the helix-induced dynamics influence the emergence of nonminimum-phase zeros.

**Reviewer:** *Additionally, in future work, it could be good to include a simulation with more than two wind turbines as a potential goal. It would be interesting to see how the balance between energy savings and blade loads changes when the number of turbines is scaled, potentially strengthening the results of this research. In general, the Future Work section of this report is very sparse, and it could be worth elaborating on a sentence for each segment of future work provided. For example, "realistic LiDAR integration" is presented, but how that would impact these results and why that is necessary is not explained clearly. There is no mention of the impact of the LiDAR modeling on the results of the study, so the potential impact of this area of future work is not clear.*

**Authors:**   We thank the reviewer for pointing out this. This section has been rewritten in the revised manuscript to provide more detailed explanations. In addition, a dedicated discussion section has been added to further elaborate on the points.

**Inserted text:** Future work can proceed along several perspectives. First, the proposed framework should be validated using a more realistic LiDAR model and in a higher fidelity simulation environment. This enables a thorough analysis and understanding of the proposed framework. Moreover, the results presented in this paper are obtained from a single simulation with one wind seed, as this study primarily aims to serve as a proof-of-concept to demonstrate the feasibility of incorporating flow information into dynamic wake mixing control. Future work should therefore consider simulations with multiple wind seeds to mitigate the influence of numerical randomness and to provide a more comprehensive characterization of wind farm behavior. To balance the power production and fatigue load, simulations with more turbine settings and different wind seeds should be conducted to evaluate the balance between energy and fatigue load. Additionally, the choice of reference signal needs to be furtherly studied to guarantee the feasibility and optimality of the wind farm performance under different wind conditions. Moreover, controllers that take the actuator and structural constraints into consideration can be designed to mitigate the increased loading on the upstream turbine. To further improve the overall performance and fully utilize the potential of the proposed framework, future works should consider conducting better feature extraction and performing quantitative flow analysis, similar to the study of [28], to directly facilitate wake mixing and better understand the influence of the proposed framework on wind flow. Finally, it is recommended to explore integrating the proposed framework with existing methods for wake mixing to enable closed-loop control. For example, combine this study with phase synchronization [27] to enable a more adaptive application.

**Reviewer:**   *Small Suggestions (line number(s) given in parentheses):*

   – *(43) citation issue*

   – *(89) WFFC used but not defined earlier, first appeared on (21)*

   – *(79) consider this capitalization: "(Light Detection And Ranging)*

   – *(79) consider replacing "remote method" with "remote sensing method."*

   – *(105) It could be beneficial to have a figure here illustrating how an MBC transform goes from a nonrotating to a fixed frame, if such a figure exists (it is difficult to conceptualize this when only looking at the equation for the linear transformation)*

   – *(115) equation three has $V_\infty$, but the following line references $U_\infty$*

   – *(183) double check axes labels for Figure 3*

   – *(225) Figure 5 should be placed much higher in the paper. Also, it should be referenced in the text or removed entirely*

   – *(276) This is a possible run-on sentence, and should be reworded*

- *(297) broken reference*
- *(325) Equation 16 has a scalar 0.4, but where this scalar comes from is not clearly established*
- *(356) should be $\mathrm{OL}_3$*
- *(375) should say "section" instead of "chapter."*
- *(386) The results of Figure 12 seem to demonstrate worse oscillations compared to open-loop operation. Consider adding an explanation for why this is the case*
- *(400) Figure 13: The bottom row has vertical grid lines, unlike the top two rows*

**Authors:** We thank the reviewer for the suggestions, and the feedback have been implemented accordingly in the text. Please note that the line numbers referenced below correspond to those in the original preprint.

- (43) We have fixed the citation issue.
- (89) We have added the definition of "WFFC" in the earlier part of the paper.
  **Inserted text :**These control approaches, known as wind farm flow control strategies (WFFC).
- (79) We have capitalized the title as the reviewer suggested.
- (79) We have changed the naming of the terminology as the reviewer suggested.
- (105) We have added Figure 2 to illustrate the pitch signal in the rotating frame and fixed frame, respectively, to help readers understand the MBC transform.
- (115) We have made sure that the inflow wind speed is denoted as $U_\infty$ in the entire paper.
- (183) We have checked the axes labels.
- (225) We have changed the position of the original figure and referred to the figure in the text.
  **Inserted text:** As a result, the overall LiDAR processing pipeline is developed as Fig.6 shows.
- (276) We have rewritten the sentence.
  **Inserted text:** To identify a model, the Pseudo-Random Binary Noise (PRBN) is selected as the excitation signal due to its effectiveness in exciting a broad spectrum of system frequencies, facilitating a comprehensive capture of the system's dynamic characteristics [13]. The magnitude of the signal is set to 1 degree. Additionally, the signal is filtered by a bandpass filter between a frequency range of $[0, 0.03]$ Hz to ensure compatibility with the actuator's bandwidth.

- (297) We have fixed the reference issue.
- (325) We thank the reviewer for pointing this out. This gain is added to scale the weight function so that control efforts are not overly penalized. We have added a sentence to explain this option.
  **Inserted text:** A scaling factor of 0.4 is adopted to ensure the controller achieves the trade-off between performance and robustness without overly penalizing control effort.
- (356) We have fixed the typo.
- (375) We have fixed the use of the word as suggested.
- (386) We appreciate the reviewer for pointing this out. As section 4.4.3 mentioned, the helix actuation is higher in this case compared to the uniform wind one. As a result, wake mixing is stronger at the downwind position [26], leading to a stronger oscillation. We have added an additional sentence in this section to explain this phenomenon.
  **Inserted text:** This facilitates better wake mixing, resulting in larger oscillations of the hub vortex.
- (400) We have replaced the plot to a one without vertical grid lines.

**References**

[1] Dewan Ahsan and Søren Pedersen. The influence of stakeholder groups in operation and maintenance services of offshore wind farms: Lesson from denmark. *Renewable Energy*, 125:819–828, 2018. doi: 10.1016/j.renene.2017.12.098.

[2] Marcus Becker, Bastian Ritter, Bart Doekemeijer, Daan van der Hoek, Ulrich Konigorski, Dries Allaerts, and Jan-Willem van Wingerden. The revised floridyn model: implementation of heterogeneous flow and the gaussian wake. *Wind Energy Science*, 2022:1–25, 2022. doi: 10.5194/wes-7-2163-2022.

[3] Marta Bertelè, Carlo L Bottasso, Stefano Cacciola, Fabiano Daher Adegas, and Sara Delport. Wind inflow observation from load harmonics. *Wind Energy Science*, 2(2):615–640, 2017. doi: 10.5194/wes-2-615-2017.

[4] Kenneth Brown and Thomas Herges. Residual uncertainty in processed line-of-sight returns from nacelle-mounted lidar due to spectral artifacts. In *Journal of Physics: Conference Series*, volume 1618, page 032052. IOP Publishing, 2020. doi: 10.1088/1742-6596/1618/3/032052.

[5] Marion Coquelet, J Gutknecht, JW Van Wingerden, Matthieu Duponcheel, and Philippe Chatelain. Dynamic individual pitch control for wake mitigation: Why does the helix handedness in the wake matter? In *Journal of Physics: Conference Series*, volume 2767, page 092084. IOP Publishing, 2024. doi: 10.1088/1742-6596/2767/9/092084.

[6] Michael Courtney, Rozenn Wagner, and Petter Lindelöw. Testing and comparison of lidars for profile and turbulence measurements in wind energy. In *IOP Conference Series: Earth and Environmental Science*, volume 1, page 012021. IOP Publishing, 2008. doi: 10.1088/1755-1315/1/1/012021.

[7] Bart M Doekemeijer, Daan van der Hoek, and Jan-Willem van Wingerden. Closed-loop model-based wind farm control using floris under time-varying inflow conditions. *Renewable Energy*, 156:719–730, 2020. doi: 10.1016/j.renene.2020.04.007.

[8] BM Doekemeijer and R Storm. Florisse_m github repository, 2019. URL `https://github.com/TUDelft-DataDrivenControl/FLORISSE_M`.

[9] BM Doekemeijer, Sjoerd Boersma, Lucy Y Pao, and Jan-Willem van Wingerden. Ensemble kalman filtering for wind field estimation in wind farms. In *2017 American Control Conference (ACC)*, pages 19–24. IEEE, 2017. doi: 10.23919/ACC.2017.7962924.

[10] Daniel Foti, Xiaolei Yang, Lian Shen, and Fotis Sotiropoulos. Effect of wind turbine nacelle on turbine wake dynamics in large wind farms. *Journal of Fluid Mechanics*, 869:1–26, 2019. doi: 10.1017/jfm.2019.206.

[11] Joeri A Frederik and Jan Willem van Wingerden. On the load impact of dynamic wind farm wake mixing strategies. *Renewable Energy*, 194:582–595, 2022. doi:

10.1016/j.renene.2022.05.110.

[12] Joeri A Frederik, Bart M Doekemeijer, Sebastiaan P Mulders, and Jan Willem van Wingerden. The helix approach: Using dynamic individual pitch control to enhance wake mixing in wind farms. *Wind Energy*, 23(8):1739–1751, 2020. doi: 10.1002/we.2513.

[13] KR Godfrey. Introduction to binary signals used in system identification. In *International Conference on Control 1991. Control'91*, pages 161–166. IET, 1991.

[14] Lars Christian Henriksen, Morten Hartvig Hansen, and Niels Kjølstad Poulsen. A simplified dynamic inflow model and its effect on the performance of free mean wind speed estimation. *Wind Energy*, 16(8):1213–1224, 2013. doi: 10.1002/we.1548.

[15] I Houtzager, JW van Wingerden, and M Verhaegen. Wind turbine load reduction by rejecting the periodic load disturbances. *Wind Energy*, 16(2):235–256, 2013. doi: 10.1002/we.547.

[16] Alan S Hsieh, Kenneth A Brown, Nathaniel B DeVelder, Thomas G Herges, Robert C Knaus, Philip J Sakievich, Lawrence C Cheung, Brent C Houchens, Myra L Blaylock, and David C Maniaci. High-fidelity wind farm simulation methodology with experimental validation. *Journal of Wind Engineering and Industrial Aerodynamics*, 218:104754, 2021. doi: 10.1016/j.jweia.2021.104754.

[17] B J Jonkman et al. Turbsim user's guide v2. 00.00. *Natl. Renew. Energy Lab*, 2014.

[18] Jason Jonkman, Sandy Butterfield, Walter Musial, and George Scott. Definition of a 5-mw reference wind turbine for offshore system development. Technical report, National Renewable Energy Lab.(NREL), Golden, CO (United States), 2009.

[19] Henry Korb, Henrik Asmuth, and Stefan Ivanell. The characteristics of helically deflected wind turbine wakes. *Journal of Fluid Mechanics*, 965:A2, 2023. doi: 10.1017/jfm.2023.390.

[20] Torben Juul Larsen, Helge A Madsen, and Kenneth Thomsen. Active load reduction using individual pitch, based on local blade flow measurements. *Wind Energy: An International Journal for Progress and Applications in Wind Power Conversion Technology*, 8(1):67–80, 2005. doi: 10.1002/we.141.

[21] Yichao Liu, Atindriyo Kusumo Pamososuryo, Riccardo MG Ferrari, and Jan-Willem van Wingerden. The immersion and invariance wind speed estimator revisited and new results. *IEEE Control Systems Letters*, 6:361–366, 2021. doi: 10.1109/LCSYS.2021.3076040.

[22] Sebastiaan Paul Mulders, Atindriyo Kusumo Pamososuryo, Gianmarco Emilio Disario, and Jan Willem van Wingerden. Analysis and optimal individual pitch control decoupling by inclusion of an azimuth offset in the multiblade coordinate

transformation. *Wind Energy*, 22(3):341–359, 2019. doi: 10.1088/1742-6596/2505/1/012006.

[23] Christian Santoni, Kenneth Carrasquillo, Isnardo Arenas-Navarro, and Stefano Leonardi. Effect of tower and nacelle on the flow past a wind turbine. *Wind Energy*, 20(12):1927–1939, 2017. doi: 10.1002/we.2130.

[24] Eric Simley, Lucy Y Pao, Rod Frehlich, Bonnie Jonkman, and Neil Kelley. Analysis of light detection and ranging wind speed measurements for wind turbine control. *Wind Energy*, 17(3):413–433, 2014. doi: 0.1002/we.1584.

[25] Richard JAM Stevens, Dennice F Gayme, and Charles Meneveau. Coupled wake boundary layer model of wind-farms. *Journal of renewable and sustainable energy*, 7(2), 2015. doi: 10.1063/1.4915287.

[26] Emanuel Taschner, Aemilius A W van Vondelen, Remco Verzijlbergh, and Jan Willem van Wingerden. On the performance of the helix wind farm control approach in the conventionally neutral atmospheric boundary layer. In *Journal of Physics: Conference Series*, volume 2505, page 012006. IOP Publishing, 2023. doi: 10.1088/1742-6596/2505/1/012006.

[27] A. A. W. van Vondelen, M. Coquelet, S. T. Navalkar, and J.-W. van Wingerden. Synchronized helix wake mixing control. *Wind Energy Science*, 10(10):2411–2433, 2025. doi: 10.5194/wes-10-2411-2025.

[28] Gopal R Yalla, Kenneth Brown, Lawrence Cheung, Dan Houck, Nathaniel de-Velder, and Nicholas Hamilton. Spectral proper orthogonal decomposition of active wake mixing dynamics in a stable atmospheric boundary layer. *Wind Energy Science Discussions*, 2025:1–36, 2025. doi: 10.5194/wes-10-2449-2025.